# Transient hypothyroidism favors oligodendrocyte generation providing functional remyelination in the adult mouse brain

Sylvie Remaud[1], Fernando C Ortiz[2,3,4†], Marine Perret-Jeanneret[1†], Marie-Stéphane Aigrot[5], Jean-David Gothié[1], Csaba Fekete[6,7], Zsuzsanna Kvárta-Papp[6], Balázs Gereben[6], Dominique Langui[5], Catherine Lubetzki[5,8], Maria Cecilia Angulo[2,3], Bernard Zalc[5], Barbara Demeneix[1*]

[1]Sorbonne Universités, Muséum d'Histoire Naturelle, Paris, France; [2]INSERM U1128, Paris, France; [3]Université Paris Descartes, Paris, France; [4]Mechanisms on Myelin Formation and Repair Lab, Instituto de Ciencias Biomédicas, Facultad de Ciencias de la Salud, Universidad Autónoma de Chile, Santiago, Chile; [5]Sorbonne Universités UPMC Univ Paris 06, Paris, France; [6]Department of Endocrine Neurobiology, Institute of Experimental Medicine, Hungarian Academy of Sciences, Budapest, Hungary; [7]Department of Medecine, Division of Endocrinology, Diabetes and Metabolism, Tupper Research Institute, Tufts Medical Center, Boston, United States; [8]AP-HP, Hôpital Pitié-Salpêtrière, Paris, France

*For correspondence:
bdem@mnhn.fr

†These authors contributed equally to this work

Competing interests: The authors declare that no competing interests exist.

**Abstract** In the adult brain, both neurons and oligodendrocytes can be generated from neural stem cells located within the Sub-Ventricular Zone (SVZ). Physiological signals regulating neuronal *versus* glial fate are largely unknown. Here we report that a thyroid hormone ($T_3$)-free window, with or without a demyelinating insult, provides a favorable environment for SVZ-derived oligodendrocyte progenitor generation. After demyelination, oligodendrocytes derived from these newly-formed progenitors provide functional remyelination, restoring normal conduction. The cellular basis for neuronal *versus* glial determination in progenitors involves asymmetric partitioning of EGFR and TRα1, expression of which favor glio- and neuro-genesis, respectively. Moreover, EGFR[+] oligodendrocyte progenitors, but not neuroblasts, express high levels of a $T_3$-inactivating deiodinase, Dio3. Thus, TRα absence with high levels of Dio3 provides double-pronged blockage of $T_3$ action during glial lineage commitment. These findings not only transform our understanding of how $T_3$ orchestrates adult brain lineage decisions, but also provide potential insight into demyelinating disorders.
DOI: https://doi.org/10.7554/eLife.29996.001

## Introduction

In Multiple Sclerosis (MS) neurological disability results from myelin and axonal degeneration. Inflammation-induced loss of myelin renders axons more susceptible to injury, leading to greater probability of permanent handicap and accumulated disease burden (*Dutta and Trapp, 2011*; *Mahad et al., 2015*). Therefore, alongside available anti-inflammatory treatments, current therapeutic strategies aim at enhancing myelin repair. In particular, screening of repurposing molecules favoring proliferation, differentiation and maturation of endogenous oligodendrocyte precursor cells (OPCs) have been deployed (*Buckley et al., 2010*; *Deshmukh et al., 2013*; *Mei et al., 2014*; *Najm et al., 2015*).

Endogenous myelin repair could exploit the two sources of OPCs that exist in the adult mammalian brain: the resident parenchymal OPCs (pOPCs) (*Dawson et al., 2003*; *Tripathi et al., 2010*; *Wolswijk and Noble, 1989*) and mouse subventricular zone-derived oligodendrocyte progenitors that then generate newly formed OPCs locally (SVZ-OPCs) (*Nait-Oumesmar et al., 2007*). Remyelination from pOPCs is known to produce thin myelin (*Xing et al., 2014*). In pathological settings such as MS, this limited remyelination produces so-called shadow plaques (*Franklin, 2002*; *Périer and Grégoire, 1965*). However, autopsies of a large cohort of MS patients show complete remyelination in certain periventricular areas, that is, in the vicinity of the SVZ, as opposed to shadow plaques, usually observed elsewhere in the brain and spinal cord (*Patrikios et al., 2006*). Furthermore, examination of post-mortem MS brains revealed mobilization of human SVZ-OPCs (*Nait-Oumesmar et al., 2007*). Together, these observations suggest that in humans, endogenous remyelination driven by SVZ-OPCs can provide full remyelination. Supporting this concept, it has been recently shown in mice that SVZ-OPCs produce thicker myelin sheets, similar to those formed during normal development (*Xing et al., 2014*). Therefore, stimulation of the regenerative potential of SVZ-OPCs could be a promising perspective for the development of therapies in the context of demyelinating diseases such as MS.

In the adult rodent, Neural Stem Cells (NSC) localized in the SVZ neurogenic niche generate glial and neuronal progenitor cells. Neuroblasts migrate tangentially toward the olfactory bulb through the rostral migratory stream (RMS) and differentiate as interneurons (*Doetsch et al., 1999*; *Lois and Alvarez-Buylla, 1993*). In contrast, oligodendrocyte progenitor derived SVZ-OPCs migrate radially and tangentially into the corpus callosum (CC), striatum or cortex and differentiate into mature myelinating oligodendrocytes. Under physiological conditions, production of SVZ-OPCs is lower compared with the number of neuroblasts that migrate along the RMS (*Menn et al., 2006*; *Suzuki and Goldman, 2003*). However, after a demyelinating insult (*Keirstead and Blakemore, 1999*; *Nait-Oumesmar et al., 1999*) or exposure to various extrinsic signals (*El Waly et al., 2014*; *Ortega et al., 2013*), NSC commitment switches towards production of oligodendrocyte progenitors that give rise to OPCs that migrate into the neighboring white matter. Stimuli favoring oligodendrocyte progenitor production include *Epidermal Growth Factor Receptor signaling (*EGFR). EGFR overexpression in the adult rodent SVZ increases oligodendrogenesis under physiological and pathological conditions, facilitating brain recovery after demyelination (*Aguirre et al., 2007*; *Doetsch et al., 2002*; *Scafidi et al., 2014*).

The continuous generation of neuronal and glial progenitors makes the adult SVZ a unique system in which to study the molecular mechanisms regulating stem cell fate decisions. Whether and how the same signal can orient cell fate decision towards one or the other direction is still an open question. Thyroid hormone (TH) is a key homeostatic signal dynamically regulating neurogenesis. Within the neurogenic niches of the adult mammalian brain, TH regulates progenitor commitment to a neuronal fate (*Kapoor et al., 2015*; *López-Juárez et al., 2012*), influencing learning, memory and mood (*Kapoor et al., 2015*; *Remaud et al., 2014*). $T_3$, the active form of TH, and its nuclear receptor TR$\alpha$1 promote NSC specification toward a neuronal identity *in vivo* through downregulation of (i) the pluripotency gene *Sox2* (*López-Juárez et al., 2012*; *Remaud et al., 2014*) and (ii) the cell cycle genes *Ccnd1* and *Myc* (*Lemkine et al., 2005*). This raises the question of whether $T_3$ influences oligodendrogenesis from NSCs, at the level of progenitor determination in the adult SVZ, upstream of its well-established role in oligodendrocyte differentiation and maturation (*Barres et al., 1994*; *Billon et al., 2002*; *Dugas et al., 2012*).

To this end, we analyzed consequences on oligodendrogenesis of $T_3$-depletion (hypothyroidism) in a well-established murine model of demyelination after which we allowed for a short period of $T_3$-stimulated remyelination. We report that in a cuprizone-induced-demyelination model (*Ludwin, 1978*) a hypothyroid window during the demyelination phase favors production of mature myelinating oligodendrocytes in the corpus callosum from SVZ-derived oligodendrocyte progenitors. This newly generated wave of SVZ-derived oligodendrocytes produces a myelin of normal thickness, whereas control euthyroid animals generate thinner myelin sheaths. By measuring compound action potentials we show complete functional recovery.

We also addressed the cellular and molecular mechanisms underlying this $T_3$-free dependent oligodendrocyte production. We show that during adult neurogenesis, the neuron *versus* glia cell fate decision in newly derived NSC progenitors is determined respectively by the presence or absence of both TR$\alpha$1 and, $T_3$, its ligand. Absence of TR$\alpha$1 in oligodendrocyte progenitors is ensured by

asymmetric segregation during mitosis of TRα1, a neurogenic factor (*López-Juárez et al., 2012*) and EGFR, an established inductor of oligodendrogenesis (*Aguirre et al., 2007*). Absence of the ligand, $T_3$, in the EGFR + oligodendrocyte progenitors is ensured by high levels of Dio3, the $T_3$ inactivating deiodinase (*St Germain et al., 2009*). This oligodendrocyte progenitor-specific Dio3 expression restricts TH availability in the glial lineage. In contrast, lower levels of Dio3 are found in neuronal progenitors where TRα1 represses *Egfr*, reinforcing neuronal commitment. Taken together, the data provide evidence that transient hypothyroidism favors lineage determination of adult multipotential SVZ-progenitors towards an oligodendroglial fate and promotes functional brain remyelination *in vivo*.

## Results

### A $T_3$-free window during demyelination favors functional remyelination

Since $T_3$ and its receptor, TRα, promote neuronal fate in the adult SVZ (*López-Juárez et al., 2012*) we reasoned that the absence of $T_3$, (hypothyroidism) might favor gliogenesis. To address this question we investigated the role of hypothyroidism in generating oligodendrocyte progenitors in the adult SVZ. Hypothyroidism was induced using MMI (0.1%) and NaClO4 (0.5%) during experimental demyelination induced by oral administration of cuprizone (CPZ) (*Figure 1A*). CPZ treatment causes marked demyelination in the corpus callosum (*Ludwin, 1978*). The demyelination phase was followed by three $T_3/T_4$ pulses to stimulate oligodendrocyte maturation. A first control was to verify responses to transient hypothyroidism without cuprizone treatment followed by $T_3/T_4$ treatment. As shown in *Figure 1—figure supplement 1A* four weeks of hypothyroidism were sufficient to stimulate significantly, and by nearly two-fold, oligodendrocyte progenitor numbers within the adult SVZ without affecting numbers of corpus callosum resident OPCs. We next quantified responses to hypothyroidism during a CPZ-induced demyelination. CPZ has the experimental advantage that the process of demyelination is temporally separated from the subsequent process of remyelination, allowing the latter to be specifically studied without the complication of ongoing demyelination (*Franklin, 2015*). CPZ-induced demyelination was accompanied by increased astrocytic (GFAP[+]) and microglial (IBA1[+]) responses (*Figure 1—figure supplement 1C,D*). In accordance with our hypothesis, transitory hypothyroidism during demyelination, followed by $T_3/T_4$ pulses (*Figure 1A*) to favor oligodendrocyte maturation (*Barres et al., 1994*), increased Myelin Basic Protein (MBP) and Myelin Oligodendrocyte (MOG), two markers of maturing oligodendrocytes, in the corpus callosum and septum (*Figure 1B* and *Figure 1—figure supplement 1B,E,F*). Moreover, transient hypothyroidism during demyelination also increased OLIG2 expression in the corpus callosum (*Figure 1—figure supplement 1G*). OLIG2 is a basic loop helix transcription factor expressed throughout oligodendrocyte lineage development (*Lu et al., 2000*). The nuclear localization of OLIG2 permitted quantification. As shown in *Figure 1—figure supplement 1G*, after 7 days of recovery, the number of OLIG2+ cells in the corpus callosum was decreased to $29.8 \pm 10.6$ OLIG2+ cells per $10^3$ $\mu m^2$ (p<0.05) in euthyroid CPZ-treated mice (EuCPZ), whereas values were not significantly different for controls (unexposed to CPZ) and hypothyroid CPZ-treated mice (HypoCPZ) ($60 \pm 8.1$ and $44 \pm 6.6$ OLIG2+ cells per $10^3$ $\mu m^2$ respectively). Interestingly, in the latter mice (HypoCPZ), we observed a restricted lateral localization of the MBP[+] and MOG[+] cell masses, above the dorsal part of lateral ventricles (see *Figure 1B* and *Figure 1—figure supplement 1B* for MBP and *Figure 1—figure supplement 1E,F* for MOG).

These mature oligodendrocytes may be generated from resident adult OPCs or they may represent newly-generated OPCs having migrated from the SVZ into the corpus callosum. To analyze their origin, we labeled progenitors within the SVZ using an eGFP lentivirus at 14d of demyelination (*Figure 1—figure supplement 2*). At 20d post-transfection by stereotaxic injection, not only did we observe GFP+ cells in the SVZ, but also groups of GFP+ cells that had migrated into the corpus callosum. We confirmed that more SVZ cells were recruited during a demyelinating insult in EuCPZ mice ($3.6 \pm 2.8$ GFP +cells per section, n = 3) compared to control mice ($2 \pm 1.4$ GFP + cells per section, n = 3). Hypothyroidism during demyelination increases this recruitment even more. HypoCPZ mice had over three times more GFP+ cells ($11.8 \pm 20.9$ GFP + cells per sections, n = 3) than EuCPZ mice, the GFP+ cells displaying characteristic oligodendrocyte morphology (*Figure 1—figure supplement 2*).

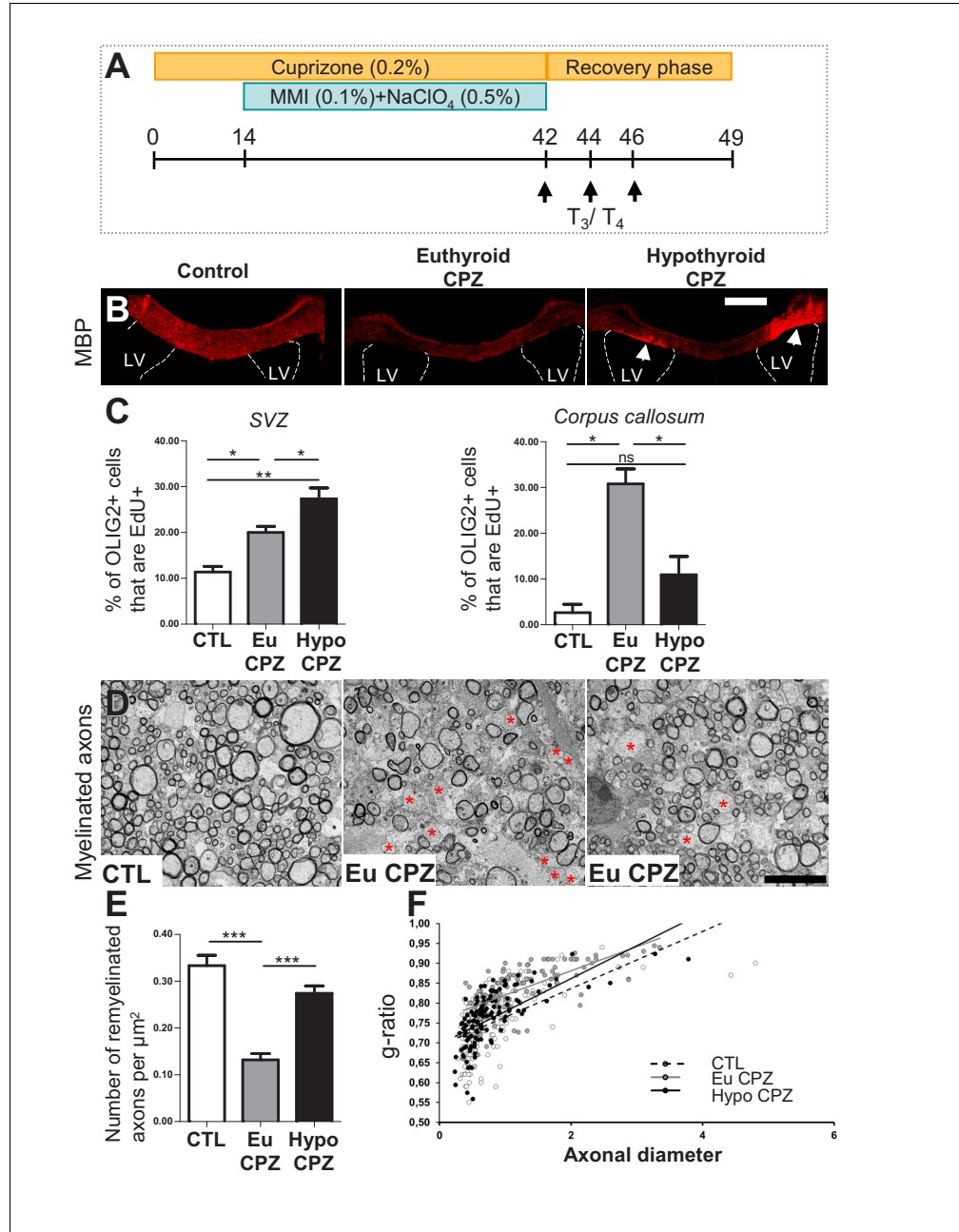

**Figure 1.** Hypothyroidism during demyelination favors oligodendrogenesis. (**A**) Experimental procedure: Cuprizone (CPZ) was added to the diet for 6 weeks (d0 to d42) and a $T_3$-free window applied for one month, two weeks after onset of CPZ treatment (d14 to d42). Three intraperitoneal injections of $T_3/T_4$ (d42, d44, d46) were given to accelerate remyelination after return to normal diet. (**B**) Extent of de/re-myelination in the corpus callosum was evaluated by MBP immunostaining 7d post-CPZ treatment. LV = lateral ventricle (**C**) Quantification of actively cycling oligodendrocyte progenitors (EdU+ OLIG2+) in the SVZ (mean ± SEM, n = 5–6 mice per group, **p=0.0015; Kruskal-Wallis test) and cycling pOPCs in corpus callosum during demyelinisation (**d28**) (mean ± SEM, n = 4–6 mice per group, *p=0.0109; Kruskal-Wallis test). (**D**) Remyelination of the corpus callosum after CPZ diet analyzed by electron microscopy of ultra-thin sections (Representative fields). Note that non-myelinated axons (red asterisks) are more numerous in EuCPZ compared to HypoCPZ samples. (**E**) Density of remyelinated axons per area (µm²) in CTL, EuCPZ and HypoCPZ after CPZ-induced demyelination (mean ± SEM, n = 12–18 sections from 2 (CTL) or 3 (EuCPZ and HypoCPZ) mice, CTL *versus* EuCPZ ***p<0.001, t = 7.952; EuCPZ *versus* HypoCPZ ***p<0.001, t = 25.835; One-way Anova test followed by Bonferroni post-hoc test) (**F**) Scatter plot of g ratio values in CTL (dotted black line, n = 198), EuCPZ (grey line, n = 112), HypoCPZ (black line, n = 122). Scale bar in (**B**): 500 µm, in (**D**): 5 µm.

*Figure 1 continued on next page*

*Figure 1 continued*

DOI: https://doi.org/10.7554/eLife.29996.002

The following source data and figure supplements are available for figure 1:

**Source data 1.** Hypothyroidism during demyelination favors oligodendrogenesis.

DOI: https://doi.org/10.7554/eLife.29996.006

**Figure supplement 1.** A $T_3$-free window during CPZ demyelination increases remyelination.

DOI: https://doi.org/10.7554/eLife.29996.003

**Figure supplement 2.** *SVZ-OPCs are recruited to the corpus callosum following a demyelinating insult.*

DOI: https://doi.org/10.7554/eLife.29996.004

**Figure supplement 3.** (A–B) Mice received $T_3/T_4$ pulses during the recuperation phase.

DOI: https://doi.org/10.7554/eLife.29996.005

Furthermore, we analyzed effects of hypothyroidism on cycling pOPCs numbers in the corpus callosum *versus* cycling SVZ-derived OPC numbers using an EdU pulse applied at day 28 of demyelination. Applying the $T_3$-free window during CPZ-induced demyelination increased cycling oligodendrocyte progenitor numbers in the SVZ, but decreased pOPC numbers in the corpus callosum (*Figure 1C*).

Taken together, these results show that the effects of hypothyroidism during demyelination preferentially stimulate SVZ-OPCs production.

MBP expression does not necessarily indicate compact myelin formation, as premyelinating oligodendrocytes also express MBP. Transmission electron microscopy (TEM) was used to quantify myelinated axons in the corpus callosum and to quantify whether the oligodendrocytes generated from hypothyroidism-induced oligodendrocyte progenitors myelinate more than resident adult pOPCs. TEM revealed increased myelinated axon density in the periventricular area of HypoCPZ *versus* EuCPZ (*Figure 1D,E*). Further, evaluation of g-ratios representing myelin thickness with respect to axon diameter revealed that the significant reduction in myelin thickness in EuCPZ (g = 0.83 ± 0.07) *versus* controls not subjected to demyelination (g = 0.75 ± 0.07, p<0.001) was rescued in HypoCPZ mice (g = 0.77 ± 0.06) (*Figure 1D,F* and *Figure 1—figure supplement 3A,B*). Both the increased myelinated axon density and myelin thickness suggest structural and functional restoration of axons in hypothyroid conditions. Moreover, we verified whether hypothyroidism applied during demyelination (and not $T_3$ pulses applied during the one-week recuperation phase) is sufficient to restore normal myelin thickness. As shown in *Figure 1—figure supplement 3C–E*, the density of myelinated axons as well as the g-ratio and the myelin thickness were also rescued in HypoCPZ compared to EuCPZ treated-mice when mice in each group only received a saline solution (NaCl) after cuprizone treatment.

To assess whether the $T_3$-free window produced functional improvement of conduction in HypoCPZ mice, we recorded evoked compound action potentials (CAPs) in coronal corpus callosum slices. The demyelinated area was recognized as a brighter region surrounding white matter (*Figure 2A* and *Figure 2—figure supplement 1*) (*Sahel et al., 2015*). In each condition tested CAPs were characterized by a fast and slow component corresponding to myelinated and unmyelinated fiber responses (*Figure 2B*). CAP amplitudes increased upon increasing pulse stimulation (*Figure 2B*), being highly variable in all groups (coefficient of variation of the first component: 67.4% in controls, 83.4% in EuCPZ, 82.4% in HypoCPZ). However, no differences were observed in normalized stimulation-response curves between groups at any stimulation pulse (*Figure 2C*; p>0.05 two way-ANOVA). Comparison of normalized CAP amplitudes revealed that the fast component was significantly reduced in EuCPZ mice compared to controls, but completely rescued in HypoCPZ mice (*Figure 2D,E*). Furthermore, the normalized area under the curve of the fast component (f-AUC) was smaller in EuCPZ mice, but regained control values in HypoCPZ mice (*Figure 2D,F*). This functional improvement was correlated with higher MBP expression levels in recorded slices of HypoCPZ mice (*Figure 2G*). These experiments indicate an extensive functional recovery of demyelinated fibers after transient hypothyroidism.

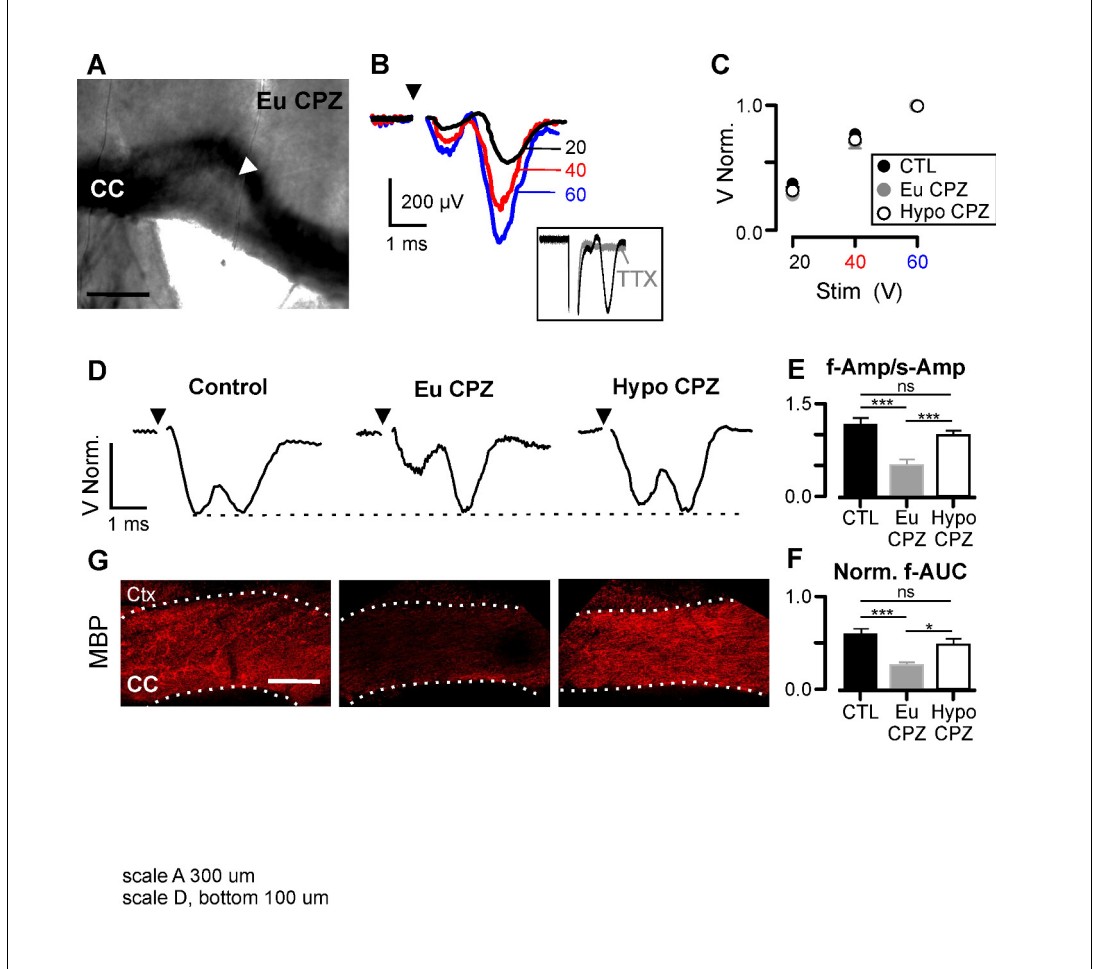

**Figure 2.** Hypothyroidism during demyelination induces functional recovery of myelinated axons. (**A**) Differential interference contrast image showing a brain slice (coronal view) from the CPZ group. CC = corpus callosum, arrowhead indicates the demyelinated lesion. (**B**) CAPs obtained following three different stimulation strengths (20, 40 and 60 V) for the brain slice shown in (**A**). CAPs were obtained after substraction of the electrical artefact recorded in presence of TTX 1 μM (inset). (**C**) CAP amplitudes normalized by the response at 60V. No differences were found between groups at any stimulation strength (n = 10–14 per group per stimulation value, p>0.05; Two-way Anova test). (**D**) Representative examples of CAPs (upper panels) for the studied groups. (**E**) Amplitude ratio between fast and slow components of the CAP (f-Amp/s-Amp, n = 10–14 per group, ***p<0.001; Kruskal-Wallis test followed by a Dunn's multiple comparison post-hoc test). (**F**) Normalized area under the curve for the fast component of the CAP (f-AUC, n = 10–14 per group, ***p<0.001, *p<0.05, Kruskal-Wallis test followed by a Dunn's multiple comparison post-hoc test). (**G**) MBP expression was assessed by immunostaining after the recordings (bottom panels), CC = corpus callosum, Ctx = cortex. Scale bars: 300 μm in (**A**), 100 μm in (**G**).

DOI: https://doi.org/10.7554/eLife.29996.007

The following source data and figure supplement are available for figure 2:

**Source data 1.** Hypothyroidism during demyelination induces functional recovery of myelinated axons.
DOI: https://doi.org/10.7554/eLife.29996.009
**Figure supplement 1.** Differential interference contrast image (coronal view) of recorded acute slices from control, EuCPZ and HypoCPZ groups.
DOI: https://doi.org/10.7554/eLife.29996.008

## Trα1 and EGFR are associated with the neuronal and glial lineages respectively

We next investigated the cellular and molecular mechanisms underlying stimulation of functional remyelination induced by a short hypothyroidwindow. As $T_3$-TRα1 is neurogenic (*López-Juárez et al., 2012*), a logical prediction would be that TRα1 should be excluded from cells expressing early oligodendroglial markers, such as OLIG2 or SOX10 (*Lu et al., 2000*; *Stolt et al., 2004*). Using the *Thra-lacZ* mouse (*Macchia et al., 2001*), we showed that TRα (βgal[+]) was strongly expressed in cells positive for DCX (*Figure 3A,D*) a neuronal lineage marker (*Sapir et al., 2000*) but

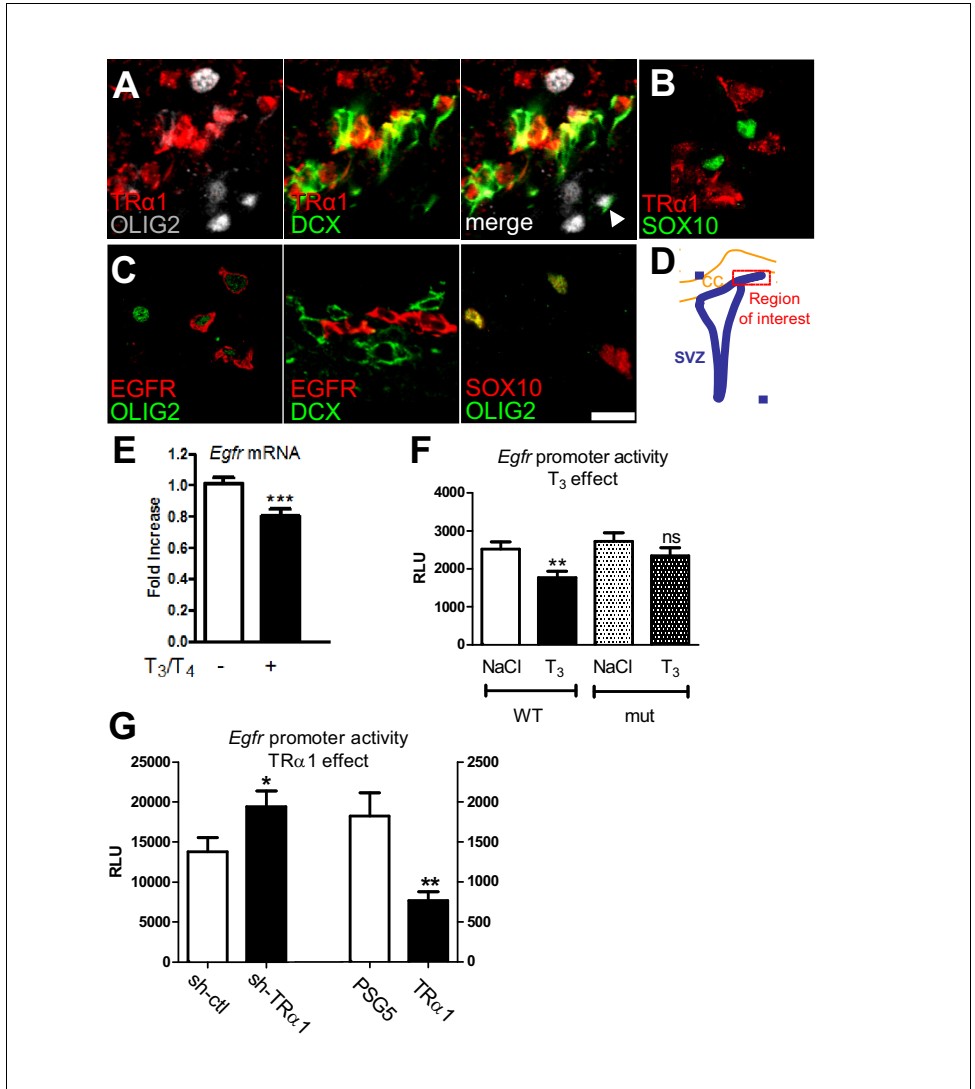

**Figure 3.** $T_3$/TRα1 signaling is excluded from the oligodendroglial lineage and represses EGFR expression in adult SVZ. (A–D) Coronal sections of adult SVZ treated for immunohistochemistry, TRα1 (red in A and B) is not expressed in OLIG2$^+$ (grey) oligodendrocyte progenitors (A), nor SOX10$^+$ (green) oligodendrocyte progenitors (B), but is expressed in DCX$^+$(green) neuroblasts (A) in the adult mouse SVZ. EGFR (red) is expressed in OLIG2$^+$ (green) oligodendrocyte progenitors (C) but not in DCX$^+$ (red) neuroblasts (middle panel in C). OLIG2 (red) and SOX10 (green) are co-expressed in oligodendrocyte progenitors (right panel in C), but very few OLIG2$^+$ cells express DCX (white arrowhead in A). (D) Schematic representation of the brain area investigated. Tagged TRα1 was detected in *TRα$^+$/°* mutant mice using a β-Gal antibody (A). To examine TRα1 expression in WT mice (B) a TRα1 antibody verified on *TRα$^+$/°* mutant mice (see materials and methods) was used. Scale bar: 10 µm. (E) $T_3$ represses *Egfr* transcription in the adult SVZ. (means ± SEM, three experiments are pooled, n = 12 mice, **p=0.001, t = 3.64, df = 30; unpaired two-tailed Student's *t* test) (F) *In vivo*, $T_3$ repression of *Egfr-luc* requires an intact TRE. Transcription from wild-type (WT) and mutated (mut) constructs was compared following saline (NaCl) or $T_3$ injection. (means ± SEM, three experiments were pooled, n = 21–24 mice per group, **p<0.01. Kruskal-Wallis test followed by a Dunn's post-hoc test) (G) Transcription from *pEgfr-Luc* following loss (*sh-TRα1 versus sh-ctl*; means ± SEM, n = 16 mice, *p=0.039, t = 2.16, df = 27; Unpaired two-tailed Student's *t* test) and gain (*TRα1* over-expression *versus* control plasmid PSG5; means ± SEM, n = 12 mice, **p=0.043, t = 3.40, df = 14; Unpaired two-tailed Student's t test with Welch's correction) of TRα1 function.

DOI: https://doi.org/10.7554/eLife.29996.010

The following source data and figure supplement are available for figure 3:

**Source data 1.** $T_3$/TRα1 signaling is excluded from the oligodendroglial lineage and represses EGFR expression in adult SVZ.

*Figure 3 continued on next page*

*Figure 3 continued*

DOI: https://doi.org/10.7554/eLife.29996.012

**Figure supplement 1.** TRα1 signalling is associated with a neuronal fate *in vitro* and *in vivo*.

DOI: https://doi.org/10.7554/eLife.29996.011

excluded from OLIG2$^+$ cells (*Figure 3A,D* and *Figure 3—figure supplement 1A*). In wild-type mice, TRα1 was also excluded from SOX10$^+$ cells, that are also OLIG2$^+$ (*Figure 3B,C*). Further, *in vitro* studies showed that TRα1 co-localized with other neuronal markers such as TUJ1 (*Figure 3—figure supplement 1B*) but could be found in more mature OPCs expressing O4, another oligodendroglial marker (*Figure 3—figure supplement 1C,C'*). As expected, EGFR, a major gliogenic factor (*Aguirre et al., 2007*), co-localized with OLIG2, but was excluded from DCX$^+$ cells (*Figure 3C,D*). These experiments show that TRα1 is expressed in neural lineage cells whereas EGFR is more restricted to glial lineage cells in the adult SVZ.

## T$_3$/TRα1 repress *Egfr* expression within the adult SVZ

The differential lineage-dependent pattern of TRα1/DCX *versus* EGFR suggested transcriptional repression of EGFR by T$_3$ in the SVZ, as seen *in vitro* (*Xu et al., 1993*). T$_3$ treatment repressed *Egfr* (*Figure 3E*) whereas hypothyroidism significantly increased SVZ *Egfr* mRNA (*Figure 3—figure supplement 1D*). In parallel, EGFR$^+$ SVZ cell numbers were significantly increased in mice lacking TRα1 (*TRα°/° mice*) (*Gauthier et al., 2001*) (*Figure 3—figure supplement 1E–G*). Lastly, using an *in vitro* neurosphere assay, we observed that T$_3$ decreased by about two-fold the numbers of EGFR + cells after 7d differentiation (*Figure 3—figure supplement 1H*). To examine T$_3$/TRα1 effects on *Egfr* transcription we used an *in vivo* gene transfer approach targeting the NSC population (*Lemkine et al., 2002*). T$_3$ repressed transcription from the wild-type *Egfr* promoter, whereas mutating a negative Thyroid Hormone Response Element (TRE) in the *Egfr* promoter abolished T$_3$/TRα1 repression (*Figure 3F*). Loss of function using sh*TRα1* increased *Egfr* promoter activity (*Figure 3G*, left) whereas gain of function with a TRα1-encoding plasmid repressed transcription (*Figure 3G*, right). Thus, *Egfr* is a negative T$_3$-regulated gene in the adult SVZ.

## Trα1 and EGFR are asymmetrically distributed in mitotic progenitors

The lineage–dependent EGFR *versus* TRα1 distributions suggested potential asymmetric partitioning during proliferation. TRα1 was found in 80% (±2%) of dividing cells, but was unequally distributed in 11% (±3.6%) of these (*Figure 4A,B*). In these, TRα1 localized to a polarized crescent in prophase (*Figure 4A,A'*), segregating preferentially to one daughter cell in telophase (*Figure 4B,B'*). Since asymmetric cell division can generate progeny with distinct fates, we analyzed respective distributions of TRα1 and EGFR. As previously reported (*Sun et al., 2005*), EGFR localized asymmetrically in mitotic cells (see *Figure 4C* for a representative telophase), forming a crescent opposite that of TRα1 during prophase (*Figure 4D,D'*). Hence, one daughter cell preferentially inherits TRα1 and the other, EGFR (see telophase, *Figure 4E,E'*). Lastly, within the specific SOX2+ mitotic progenitor population, asymmetric TRα1 localization was observed in 20% (±3.4%) of these SOX2 + cells (*Figure 4F*). This smaller proportion of cells displaying asymmetric TRα1 matches the oligodendrocyte *versus* neuron ratio produced in the adult SVZ in normal conditions, with the large majority of cells being neuronal (*Menn et al., 2006*).

## SVZ oligodendrocyte progenitors are protected from T$_3$ action by type three deiodinase expression

Transcriptional activity of TRα1 is modulated by availability of T$_3$, the biologically active form of thyroid hormone. T$_3$ availability is determined by the opposing actions of activating and inactivating deiodinases (type 2 (Dio2) and type 3 (Dio3) respectively) (*Gereben et al., 2008*). As T$_3$/TRα1 together downregulate *Egfr* we hypothesized that T$_3$ availability could be limited by cell-specific distribution of the inactivating deiodinase Dio3, especially in EGFR$^+$ activated NSC and progenitors. Dio3 was strongly expressed in SVZ-NSCs co-expressing Nestin and SOX2 (*Figure 5A*) or GFAP (*Figure 5—figure supplement 1A*). Furthermore, Dio3 expression was strongest in EGFR$^+$ (*Figure 5B* and *Figure 5—figure supplement 1B*) and SOX2$^+$ (*Figure 5—figure supplement 1C,D*) progenitors

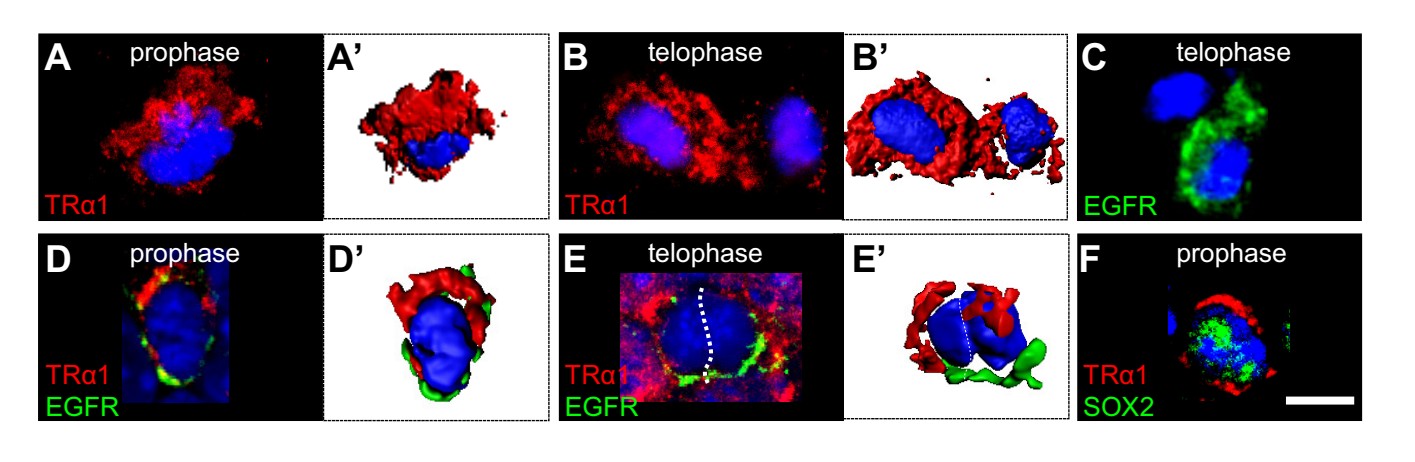

**Figure 4.** Asymmetric TRα1/EGFR segregation in the adult SVZ. (A–F) TRα1 and EGFR segregate asymmetrically in the adult SVZ *in vivo*. Coronal sections of WT adult mouse brain across the lateral ventricles were immunostained with anti-TRα1 (red) and/or anti-EGFR (C-E', green) or anti-SOX2 (F, green) antibodies. Mitotic figures in the SVZ were identified under confocal microscopy (A–F) and 3D reconstructions (A', B', D', E') performed. (A and A') During prophase, TRα1 is preferentially localized at one pole of the dividing cell. (B and B') In telophase, only one of the daughters (left cell) has inherited largely TRα1. The other (on the right) shows residual TRα1 expression. (C) During telophase, only one daughter cell (lower cell) inherited EGFR. (D and D') Double labeling for TRα1 and EGFR shows opposite pole localizations during prophase. (E and E') During telophase, the left daughter is enriched in TRα1 whereas the cell on the right received predominately EGFR. (F) Asymmetric distribution of TRα1 (upper pole) occurs in SOX2[+] progenitors Scale bar: 5 µm.

DOI: https://doi.org/10.7554/eLife.29996.013

and absent from DCX[+] neuroblasts (*Figure 5B* and *Figure 5—figure supplement 1B*). Thus, even if EGFR[+] cells express low levels of TRα1, higher Dio3 will protect them from the neuronalizing effects of $T_3$. In the *Pdgfrα:GFP* mouse (*Hamilton et al., 2003*), PDGFRα[+] OPCs were still Dio3 positive (*Figure 5C*). Strong co-expression of EGFR/Dio3 was confirmed *in vitro*, using adult SVZ neurosphere cultures: in basal conditions, DCX[+] cells had barely detectable Dio3 as opposed to strongly Dio3 expressing-EGFR[+] cells (*Figure 5—figure supplement 1E*).

To functionally address the significance of Dio3 expression in EGFR[+] cells, we examined effects of loss of Dio3 function, using gene transfer *in vivo* with sh*Dio3* expressing-plasmid. Down-regulating Dio3 expression by sh*Dio3* significantly reduced cell proliferation by 50% (sh*ctl*: 9.2 ± 0.6 versus sh*Dio3*: 4.6 ± 0.8, p=0.0097, *Figure 5D*) and numbers of OLIG2[+]oligodendrocyte progenitors by 30% (sh*ctl*: 55.5 ± 5.8 versus sh*Dio3*: 35 ± 6.6, p=0.032, *Figure 5E*), showing that down-regulation of *Dio3* is sufficient to modulate oligodendrogenesis.

To address further the effects of variations in $T_3$ availability on cell fate, we used neurosphere cultures. A 24 h, $T_3$ (1 nM to 50 nM) pulse added before cell differentiation, promoted DCX[+] neuronal genesis at the expense of OLIG2[+] populations. $T_3$ (1 nM) decreased five-fold the ratio of OLIG2[+]-*versus* DCX[+] cell numbers (*Figure 5—figure supplement 2A,B*).

Bringing together these *in vivo* and *in vitro* results shows that type three deiodinase, preferentially expressed in oligodendrocyte progenitors, maintains oligodendrogenesis in the adult SVZ.

## Discussion

These results show that transient hypothyroidism stimulates oligodendrocyte production, by favoring oligodendrocyte progenitor lineage decisions (*Figure 6*). We demonstrated that a $T_3$-free window promoted SVZ-derived oligodendrocyte progenitor and OPC generation followed by functional remyelination in response to a demyelinating insult. In accordance with others studies SVZ-OPC-driven remyelination, as opposed to resident adult pOPC-derived remyelination, resulted in wrapping of denuded axons with myelin of normal thickness, restoring functional nerve conduction. Our results strongly suggest that SVZ-derived OPCs are more prone to participate to efficient brain remyelination than resident OPCs.

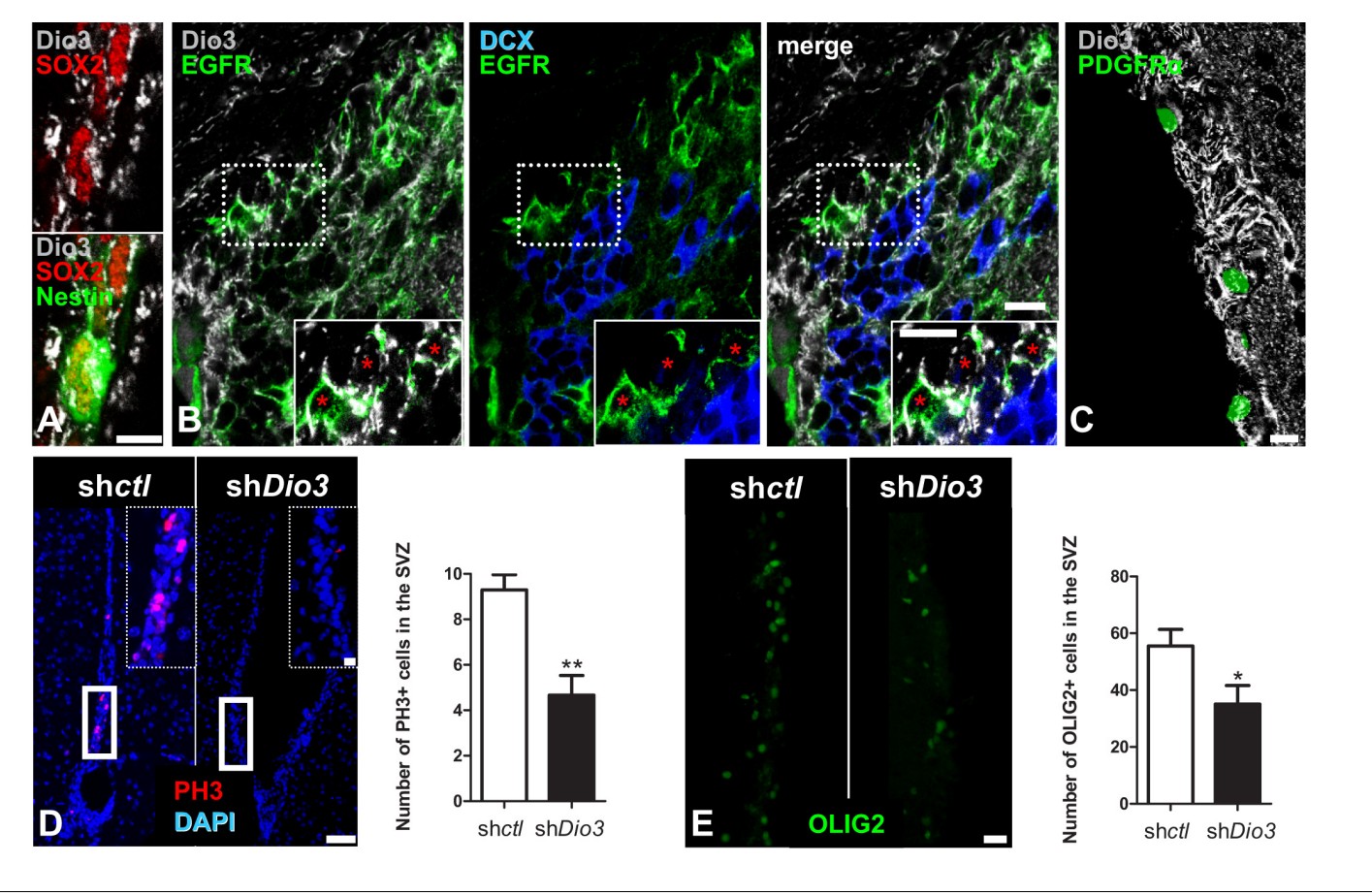

**Figure 5.** Cell-specific Dio3 expression in the adult SVZ. (**A–C**) The inactivating deiodinase, Dio3, is detected in SVZ-derived OPCs, but not in DCX[+] neuroblasts. (**A**) Dio3 (grey) is detected in NSC identified by co-expression of SOX2 (red) and Nestin (green) markers. (**B**) EGFR[+] progenitors (green) express high levels of Dio3 (grey). In contrast, Dio3 is hardly or not detected in DCX[+] neuroblasts (blue). Dotted squares denote regions magnified in insets. Red asterisks denote EGFR+ progenitors expressing high levels of Dio3. (**C**) In the *Pdgfrα-H2B:GFP* transgenic adult mouse (nuclear GFP staining), Dio3 (grey) is highly expressed in SVZ-derived OPCs (GFP[+] cells). (**D**) Immunostaining against PH3 (mitotic marker, red) and (**E**) OLIG2 (OPC marker, green) following stereotaxic injection into the SVZ of *shDio3* or *sh-control* (*shctl*) plasmids. Dotted squares denote regions magnified in insets. Quantification of PH3[+] (mean ± SEM, two experiments were pooled, n = 12 slides from 6 animals, **p=0.0097, U = 11, Mann Whitney test) and OLIG2[+] (mean ± SEM, two experiments were pooled, n = 10 slides from 6 animals, *p=0.0325, t = 2.316, df = 18; Unpaired two-tailed Student's *t* test) cells in the SVZ in the injected hemisphere. Scale bar in A-C: 10 μm, in D and E: 20 μm (inset, 50 μm).

DOI: https://doi.org/10.7554/eLife.29996.014

The following source data and figure supplements are available for figure 5:

**Source data 1.** Cell-specific Dio3 expression in the adult SVZ.
DOI: https://doi.org/10.7554/eLife.29996.017
**Figure supplement 1.** Expression of the inactivating deiodinase, Dio3, excludes TH signalling from NSC, progenitors and oligodendrocyte progenitors.
DOI: https://doi.org/10.7554/eLife.29996.015
**Figure supplement 2.** (A and B) SVZ tissues were sampled from WT adult mice for neurosphere culture.
DOI: https://doi.org/10.7554/eLife.29996.016

Under euthyroid conditions, oligodendrocyte progenitor determination is favored by three factors that establish a $T_3$/TRα1-free state around SVZ progenitors. First, there is asymmetric distribution of EGFR and TRα1 in a subpopulation of proliferating progenitors. Second, the presence of the inactivating deiodinase, Dio3 in NSCs, progenitors and early OPCs will ensure degradation of any surrounding $T_3$. Third, this Dio3-generated absence of $T_3$ will limit repression of EGFR by any residual TRα1. Together these mechanisms block $T_3$/TRα1 signaling in glial-restricted progenitors. In contrast, in cells receiving TRα1, residual *Egfr* expression will be repressed by $T_3$/TRα1 permitting neuronal determination (*López-Juárez et al., 2012*).

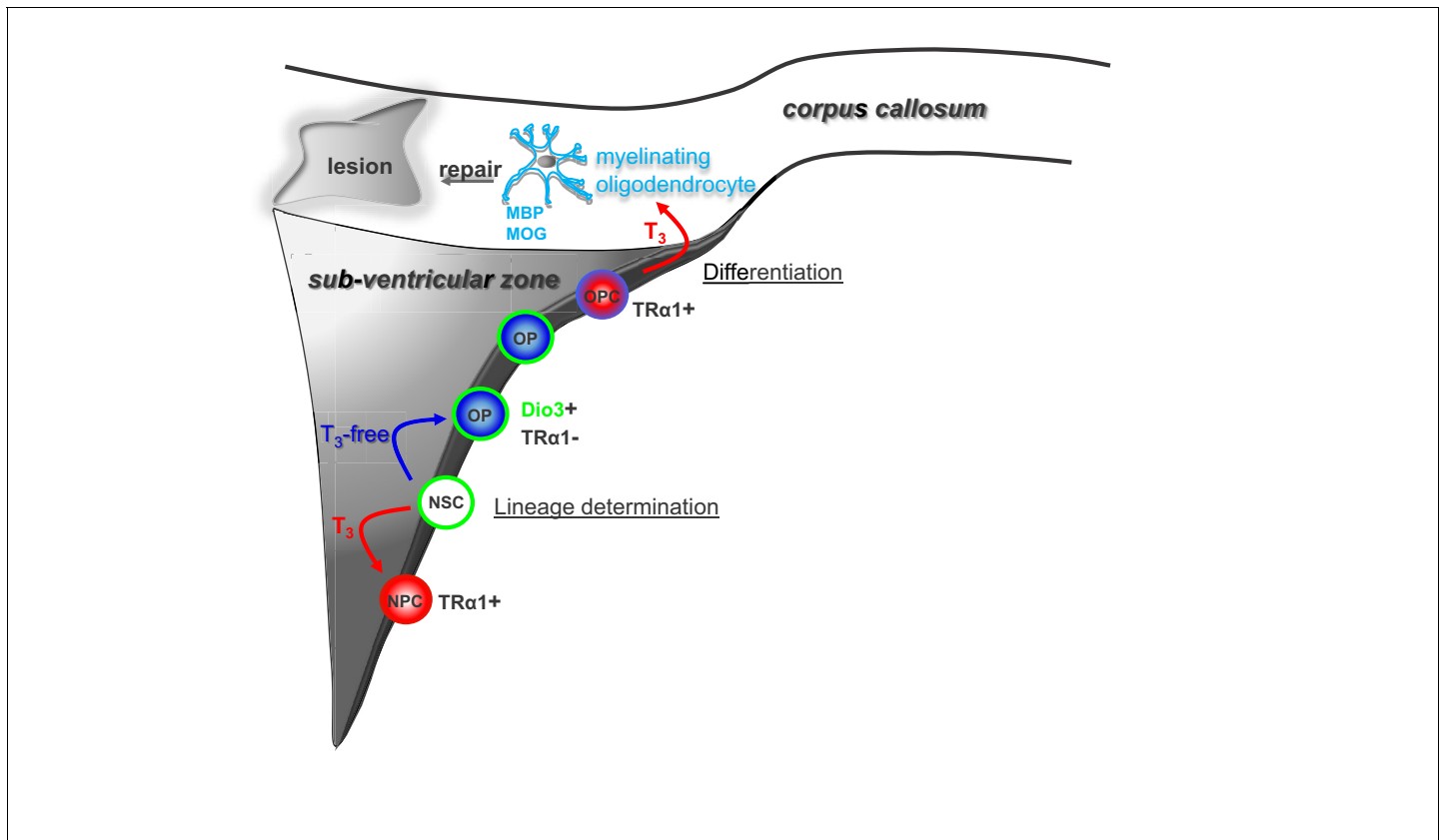

**Figure 6.** Modulation of timing of thyroid hormone (T$_3$) availability in the adult brain modifies cell fate decisions (neuronal *versus* oligodendroglial fate choice) in the sub-ventricular zone (SVZ). T$_3$ through its nuclear receptor, TRα1, favours progenitor commitment toward a neuroblast phenotype. In contrast, a transient lack of T$_3$ (T$_3$-free window) increases numbers of oligodendrocyte progenitors (OPs) and thus SVZ-derived OPCs with the capacity to repair myelin within the corpus callosum (CC) and to restore nerve conduction after a demyelinating insult. Note that, following progenitor asymmetric division, the oligodendrocyte progenitor is protected from the neuralizing effects of T$_3$ by (i) expression of the gliogenic factor EGFR and the T$_3$-inactivating enzyme (Dio3) and by (ii) the exclusion of TRα1.

DOI: https://doi.org/10.7554/eLife.29996.018

These results could appear paradoxical given the rich datasets detailing the importance of T$_3$ for oligodendrocyte differentiation (*Barres et al., 1994*; *Billon et al., 2002*), maturation and myelination (*Dugas et al., 2012*). Yet our results do not contradict this well-established concept. As expected, addition of T$_3$ following a period of hypothyroidism consolidates remyelination (*Figure 1*). Strikingly, our data indicate that transient hypothyroidism during demyelination, prior to the addition of T$_3$, activates oligodendrogenesis in the dorsal SVZ favoring migration of remyelinating OPCs into the corpus callosum.

Even if we cannot exclude a role of resident adult pOPCs following a transient lack of T$_3$, several arguments from our experiments favor a facilitating role of SVZ-derived OPCs in functional brain repair. First, cell tracing using a eGFP lentivirus showed that the remyelinating OPCs derive from the SVZ. Second, a hypothyroid window (without cuprizone treatment) was sufficient to bolster significantly SVZ-OPCs and not resident pOPCs. Lastly, transient hypothyroidism increased the number of cycling oligodendrocyte progenitors in the SVZ, but decreased the number of cycling pOPCs located within the corpus callosum. Each of these findings suggest that SVZ-derived oligodendrocyte progenitors preferentially respond to a lack of T$_3$, as compared to pOPC populations.

The efficient remyelination capacity of SVZ–mobilized OPCs has been well-documented experimentally in the mouse brain (*Nait-Oumesmar et al., 2007*; *Xing et al., 2014*). In accordance with our results, *Xing et al. (2014)* and *Brousse et al. (2015)* demonstrated by *in vivo* fate-mapping that the region of the corpus callosum adjacent to the SVZ is more permissive to remyelination than that arising from pOPCs. A higher density of SVZ-derived OPCs is found adjacent to the dorso-lateral

part of the SVZ after acute demyelination (*Xing et al., 2014*; Brousse et al., 2015). Moreover, our study of myelin integrity deposited by newly generated oligodendrocytes after one-week of recovery in hypothyroid mice showed a g-ratio similar to control mice and significantly lower than in euthyroid mice. Of note, since animals were not examined for longer periods after stopping the cuprizone diet, we cannot ascertain the stability of the remyelination observed. However, this finding is again in agreement with published data showing that axons myelinated by SVZ-derived oligodendrocytes have lower g-ratios than axons myelinated by pOPCs (*Xing et al., 2014*). Taken together, our work and previous studies (*Xing et al., 2014*); Brousse et al., 2015) show that resident pOPCs should not be considered as the only source of remyelinating cells and that SVZ-OPCs are also involved in efficient myelin repair.

A potential caveat might be that our observation is limited to the corpus callosum and that our findings cannot be extrapolated to other brain or spinal cord regions. However, in MS, lesions of the corpus callosum and prefrontal cortex are also of primary clinical concern, with ensuing cognitive impairments affecting decision-making with crucial consequences. So these findings have clinical relevance, although we realize that reaching therapeutic applications through local modulation of thyroid hormone availability is complicated by the fact that MS lesions are asynchronous and not localized to one particular area.

Turning to the capacity of resident OPCs to carry out remyelination, it is plausible that, despite the thin myelin and short internodes generated, conduction speed could be partially restored. A potential scenario would be recovery from a complete conduction block (demyelinated) to 45 m/s (thin myelin) as opposed to 50 m/s with thick myelin. However, cognitive actions depend on rapid decision-making and so, even in daily life, this slower conduction could still represent a severe handicap. In this respect, the recovery of CAPs suggests that remyelination in the cuprizone model preserves axon integrity as recently shown in an EAE model (*Mei et al., 2016*).

It is interesting to note that, although the movement or mobilisation of cells can only be inferred when examining post-mortem tissue, in brains of MS patients, early glial progenitors were observed more frequently in periventricular lesions as opposed to lesions further away from the ventricle (*Nait-Oumesmar et al., 2007*). Furthermore, in experimental models (*Bunge et al., 1961*; *Stidworthy et al., 2003*) as well as in MS patients (*Périer and Grégoire, 1965*), lesions that spontaneously remyelinate, forming 'shadow plaques', are characterized by a thinner myelin sheath. From our data, one could predict that remyelination closest to the SVZ should not yield shadow plaques because oligodendrocytes generated from SVZ-derived OPCs myelinate more efficiently and with normal sheath thickness. Indeed, *Patrikios et al. (2006)* described 51 autopsies of human MS cases and observed that complete remyelination was detected in some periventricular plaques, contrasting with shadow plaques observed at a greater distance from ventricles. This finding bolsters the clinical relevance of our findings and strengthens the concept that transient hypothyroidism could enhance generation of SVZ-derived OPCs in demyelinating diseases, a property that could be exploited for translational purposes.

Here we show that nerve conduction measured by evoked compound action potential is altered in EU-CPZ treated-mice and is completely restored in Hypo-CPZ animals. Crawford and collaborators reported that following 6 weeks-cuprizone diet with 3 to 6 weeks of normal diet led to regeneration of myelin but did not fully reverse nerve conduction (*Crawford et al., 2009*), underlining the contrast with our data in hypothyroid conditions. Furthermore, using a mouse model of hypoxia, *Scafidi et al. (2014)* demonstrated that EGFR treatment rescued the increase in g-ratio in association with a complete functional recovery. These data emphasize the importance of myelin repair to recover normal functional axon conduction.

Our work demonstrates how an inactivating deiodinase that limits availability of a homeostatic endocrine signal, $T_3$, determines neuronal cell fate decision. This result has evolutionary implications. Beyond the questions raised by co-evolution of the thyroid gland and the appearance of myelination, current hypotheses on the changes in vertebrate brain structure and function implicate control of progenitor pool size (*Miller et al., 2014*). Interestingly, it has been shown that the Drosophila EGFR homologue DER regulates midline glial cells survival during embryonic and larval stages in the fly (*Bergmann et al., 2002*; *Hidalgo et al., 2001*; *Zak and Shilo, 1990*). The steroid ecdysone receptor (EcR), the invertebrate homolog of TR that controls metamorphosis in insects, stops mitotic activity in the glial population and may induce terminal differentiation by interfering with Ras/MAPK activity underlying DER activation (*Giesen et al., 2003*). Thus, the ecdysone receptor may be a

functional homologue of vertebrate TR in the regulation of the number of glial cells and their differentiation. It is well established that brain development shares remarkable similarities between vertebrates and invertebrates, notably in the biology of NSC niches (*Reichert, 2009*). Here we propose a new evolutionary conserved crosstalk, between EGFR and the nuclear receptor superfamily containing EcR and TR, between vertebrates and invertebrates in the regulation of gliogenesis.

In conclusion, three major findings arise from this work. First, our results place a key homeostatic factor, thyroid hormone, at the crossroads of cellular decisions in the adult mouse neural stem cell niche, modulating actions of other players including TRα1 or EGFR that determine adult NSC commitment to neuronal or oligodendrocyte progenitors, respectively. Second, we observe that newly-generated OPCs produce myelin of normal thickness. This contrasts with the thinner myelin sheath usually observed when remyelination is carried out by resident pOPCs, a situation that leads in MS to shadow plaques. Finally, remyelination with a thick myelin sheath provides a fully functional recovery, suggesting novel therapeutic targets for demyelinating disease.

## Materials and methods

### Animals
Swiss and C57BL/6 wild-type male mice, 8 weeks-old, were purchased from Janvier (Le Genest St. Isle, France). Food and water were available ad libitum. All procedures were conducted according to the principles and procedures in Guidelines for Care and Use of Laboratory Animals. $TR\alpha^{\circ/\circ}$ mice (*Gauthier et al., 2001*), lacking all TRα gene products, were maintained on a 129/Sv strain (wild-type 129/Sv mice were purchased from Charles River Laboratories, L'Arbresle, France). TR gene was disrupted with an in-frame insertion of the *LacZ* reporter gene. The *Thra-LacZ* mouse was obtained from Karine Gauthier (ENS, Lyon France).

Two-month-old PDGFRα$^{EGFP}$ mice were purchased from Jackson Laboratory (RRID:IMSR_JAX: 007669). They express a H2B-eGFP fusion protein from the endogenous *Pdgfrα* locus.

### Hypothyroid and hyperthyroid treatments
To induce hypothyroidism, 8 week-old male mice were given iodine-deficient food containing 6-n-propyl-2-thiouracil (PTU) at 0.15% (Harlan Tekland, Madison, WI) for two weeks. To induce hypothyroidism in newborn mice, pregnant dams were fed with PTU diet from day 14 of gestation through lactation. To test TH effects, $T_3$ alone (2.5 µg/g body weight, bw) diluted in physiological solution (0.9% saline) was injected subcutaneously (s.c) in newborn mice or a mix of $T_3$/$T_4$ (0.3/0.012 µg/g bw) was given intraperitoneally (i.p.) in adult mice along with 1.2 µg/ml of $T_4$ in drinking water. Furthermore, the hypothyroid window in demyelination experiments (see below) was induced by administrating 0.25% methimazole (MMI) and NaClO4 (0.5%) in drinking water for 28 days.

### Plasmids, non-viral injections for *in vivo* studies
*Egfr-luc* plasmid, provided by *Zhang et al. (2007)*, contains −1242 to −21 bp of the mouse *Egfr* promoter cloned upstream of the Firefly luciferase coding sequence of the pGL3-enhancer *Luciferase* reporter vector. One putative *TRE* in the *Egfr* promoter region was mutated in the *Egfr-luc* reporter plasmid using the QuickChange II XL Site-directed Mutagenesis kit (Strategene, La Jolla, CA). The plasmid driving transcription of a *shRNA* against *Dio3* (*sh-Dio3*) was generated by Eurofins MWG Operon. The *sh-Dio3* was produced by gene synthesis and incorporated downstream of a *CMV-H1* hybrid promoter (*Hassani et al., 2007*). The *sh-TRα1* and the TRα1 expressing plasmids were as described (*Hassani et al., 2007*).

Plasmids DNA were diluted in a 5% glucose solution and complexed with in vivo-jetPEI (Polyplus transfection) with a ratio of 6 PEI nitrogen per DNA phosphate (*Goula et al., 1998*). As previously described (*Goula et al., 1998*), one-day old newborn mice were stereotaxically transfected into the lateral ventricle of the brain (bilateral injection) with 1 µg DNA in 2 µl. Mice were sacrificed 48 hr after transfection and SVZ were dissected and luciferase activity was assayed following the manufacturer's protocol (Promega, Madison, Winsconsin, USA). Luciferase activity was measured 48 hr after transfection to allow *shRNA* expression. Adult Swiss OF1 mice were transfected by stereotaxic injection into one lateral ventricle (unilateral injection, 0.2 mm posterior to bregma, 1.1 mm lateral, 2.2 mm deep from the pial surface) with 2.5 µg DNA (maximal dose) in 5 µl.

## Lentiviral injection

SVZ-cells were labeled using a CMV-eGFP lentivirus (generously provided by Marie-Stéphanie Aigrot, ICM, Paris) by stereotaxic injection. One µl of the lentivirus stock, titered as 1.64E + 09 TU/ml by FACS, was injected into the lateral ventricles of adult mice (n = 3 mice per group) using the same coordinates as described above. Viral particles were injected over 10 min at 300 nl/min to limit reflux along the needle track. We also used an ultra-fine silica fiber needle (200 × 100 µm) that reduces damage during intra-cerebro-ventricular injections.

## Immunostaining

Mice were anesthetized with Pentobarbital (130 mg/kg, Centravet) and perfused through the left heart ventricle rapidly with PBS and then with 4% paraformaldehyde in PBS (0.1 M, pH 7.4) and post-fixed at room temperature for 2 hr in the same fixative solution. Brains were cryoprotected in 30% sucrose in PBS at 4°C, embedded in OCT, frozen and stored at −80°C until processed. Brain coronal sections (30 µm thick) were incubated for 1 hr in a blocking solution of 10% normal donkey serum (Sigma) and 1% BSA (Sigma) in PBS and then incubated with the primary antibody diluted in the blocking solution overnight at 4°C. After washing in PBS (3 times for 10 min), sections were incubated with fluorescent secondary antibodies (1/500, Invitrogen) for 2 hr at room temperature. Three washing in PBS were then performed and sections were mounted onto SuperFrost glass slide (Fisher) before being mounted with Prolong Gold antifade reagent containing DAPI (Invitrogen).

Primary antibodies used: rabbit anti-TRα1 (Rockland, 1/300, see below for antibody specificity verification), rabbit anti-EGFR (Millipore, 1/300), rabbit anti-Sox2 (Millipore, 1/300), goat anti-Sox2 (Santa-Cruz, 1/200), rabbit anti-PH3 (Millipore, 1/300), rabbit anti-Olig2 (Millipore, 1/300), guinea-pig anti-DCX (Millipore, 1/300), guinea-pig anti-Sox10 (gift from M. Wegner, 1/600), rabbit anti-Dio3 (Novus Biologicals, 1/500), rabbit anti-MBP (Millipore, 1/500), chicken anti-GFAP (Millipore, 1/500), rabbit anti-Iba1 (Millipore, 1/500), mouse anti-O4 (cultured supernatant produced in the lab, 1/10), rabbit anti-Sirt1 (Millipore, 1/500), mouse anti-Tuj1 (Abcam, 1/200), goat anti-β-galactosidase (Cappel, 1/300).

The specificity of the TRα1 antibody (Rockland) was determined in three distinct experimental settings. First, immunocytochemistry with this antibody gave exactly the same expression pattern as with a βgal-antibody used in the heterozygous *TRα/LacZ* mouse. In this mouse, the fusion protein TRα-βgal, is expressed from the TRα allele (*Macchia et al., 2001*) as reported in *López-Juárez et al. (2012)*. Second, DCX was found to be co-expressed with TRα1 when TRα1 was detected either using Rockland antibody in WT mice (as in *López-Juárez et al., 2012*) as well as with βgal antibody to detect tagged TR protein in *TRα/LacZ* mice (*Figure 3A*, this paper). Finally, the antibody gave the same cell-specific pattern as GFP following in vitro transfection of pTRα1:GFP DNA plasmid (*Mavinakere et al., 2012*) transfection using the neurosphere assay (data not shown).,

Secondary antibodies used: donkey anti-rabbit (Alexa Fluor 488, 594 and 647 nm), donkey anti-guinea pig (594 nm), donkey anti-goat (594 nm), donkey anti-chicken (594 nm), donkey anti-mouse (594 nm) all used diluted (1/500) in PBS containing 1% BSA and 1% normal donkey serum.

Images were acquired using LSM 710 and SP5 confocal microscopes. For confocal imaging of asymmetric cell division, a stack of images through the SVZ (a minimum of 20 images with a 0.2 µm step in the z-axis) was collected and projection (average) images made from z-stacks using Fiji software Amira software was used to create 3D reconstructions of the asymmetric cells. Note that cell divisions were easily observed using two parameters: (i) PH3 staining reporting cell undergoing mitosis and (ii) condensed chromatin throughout the various stages of mitosis. The major phases of cell cycle progression were determined as a function of chromatin condensation, nuclear size and separation of daughter nuclei.

A signal threshold was applied using Fiji software to separate cell populations expressing different levels of protein markers determined by immunohistochemistry. Following this analysis, the statistical validity of the differences between sub-populations was established using ANOVA test.

## Real-time PCR analysis

Lateral walls of the lateral ventricles were dissected under a binocular microscope, snap frozen in liquid nitrogen and stored at −80°C. Total RNA was extracted with RNAble according the protocol provided with the reagent (Eurobio), concentrations/integrity were measured and stored at −80°C in

Tris 10 nM (ou mM)/EDTA 0.1 mM (pH 7.4). One microgram of total RNA was reverse-transcribed using high capacity cDNA reverse Transcription kit (Applied Biosystems). For qPCR analysis, 2 µl of cDNA (diluted 20-fold) were used as template in a final volume of 20 µl per well. TaqMan specific probes (Applied Biosystems) used were the following: *Egfr* (Mm00433023_m1) and control assays (GAPDH, Mm99999915_g1). qPCR reaction was performed using ABI PRISM 7300 Sequence Detection System (Applied Biosystems) as described previously (*Decherf et al., 2010*).

## Neurosphere culture and differentiation

Five animals were sacrificed per preparation and the brains were sliced into small pieces. SVZ were dissected in DMEM-F12-glutamax (Gibco) and the lateral walls of the lateral ventricles were incubated in papain solution for 30 min at 37°C to obtain a single-cell suspension. Cells were collected by centrifugation at 1000 rcf for 5 min. The cellular pellets were resuspended in 200 µl of DMEM-F12 (Gibco). Cells ($10^5$) were cultured in complete culture medium (DMEM-F12 [Gibco], 40 µg/ml insulin [Sigma], 1/200 B-27 supplement [Gibco], 1/100 N-2 supplement [Gibco], 0.3% glucose, 5 nM Hepes, 100 U/ml penicillin/streptomycin) containing 20 ng/ml of EGF and 20 ng/ml of FGF2 (Peprotech) for one week to obtain primary neurospheres. The cultures were maintained in a 5% $CO_2$ environment at 37°C for 7 days. To analyse cell differentiation, neurospheres were dissociated into single cells and $2.10^5$ cells were plated on Matrigel-coated glass coverslips in 24-well plated in complete culture medium without EGF/FGF2 for 7 days. To test the effects of $T_3$ on the early neuron/glia cell fate choice, a dose-dependent pulse of $T_3$ (1 nM, 10 nM, 50 nM) was given at day 1 of differentiation. Cells from the control group were grown in complete culture medium with no addition of $T_3$. Six days later, cells were fixed with 4% paraformaldehyde for 10 min.

## Demyelination induction

Adult male mice C57BL/6, aged 8 weeks, were fed with 0.2% cuprizone (w/w) mixed into standard mouse chow for a total of 6 weeks (42 days) to induce demyelination in the corpus callosum. After cuprizone treatment, the diet was changed to normal chow for one week before studying remyelination at 49d. Aged matched normal controls (CTL) were fed with a standard chow without cuprizone. To analyze the effects of a lack of $T_3$ on remyelination, a critical hypothyroid window was applied during the active phase of demyelination, for one month (from 14d to 42d).

## Transmission Electron Microscopy

One week after the end of the CPZ diet animals were perfused with an intra-cardiac injection of a solution containing 1.6% glutaraldehyde in phosphate buffer (0.12M, pH7.4). Brains were dissected and post fixed overnight in the same fixative solution at +4°C. Thick coronal sections (1 mm) at the level of the lateral ventricle (mouse brain atlas from Paxinos interaural: 4.30 mm bregma: 0.50 mm) were performed using a mouse brain slicer matrix (Zivic Instruments). Sections were rinsed with PB and post fixed with 1% osmium tetroxide in PB for 1 hr. Samples were dehydrated in a graded series of ethanol solutions (75, 80, 90% and 100%, 10 min each). Final dehydration was performed twice in 100% acetone for 20 min. Infiltration with Epon812 (epoxy resin) was performed in 2 steps: 1 night at +4°C in a 50:50 mixture of Epon and acetone, 6 hr at RT in fresh Epon812. Half sections were flat embedded with a drop of fresh resin in between two pieces of Aclar plastic foil. Polymerization was performed at 56°C for 48 hr in a dry oven. For sectioning samples were positioned at 90° (from flat) and cut with an UC7 ultramicrotome (Leica). Semi-thin sections (0.5 µm thick) were stained with 1% toluidine blue in 1% borax. Ultra-thin sections (70 nm thick) were contrasted with 2% uranyl acetate and Reynold's lead citrate (Reynolds, ES (1963). The use of lead citrate at high pH as an electron opaque stain in electron microscopy. They were observed with a Hitachi HT7700 electron microscope, operating at 80 kV. Pictures (2048x2048 pixels) were taken with an AMT41B (pixel size: 7.4 µm x7.4 µm).

## Acute slice preparation and compound action potential recordings

Acute coronal slices of *corpus callosum* (300 µm) were prepared from 3.5-month-old mice in an ice-cold solution containing (in mM): 93 N-methyl-d-glucamine, 2.5 KCl, 1.2 $NaH_2PO_4$, 30 $NaHCO_3$, 20 HEPES, 25 glucose, 2 thiourea, 5 ascorbate, 3 pyruvate, 0.5 $CaCl_2$, and 10 $MgCl_2$ and incubated for 8 min in the same solution at 34°C (pH 7.4 equilibrated with HCl). Then, they were transferred for 20

min at 34°C into a standard extracellular solution containing (in mM): 126 NaCl, 2.5 KCl, 1.25 NaH2PO4, 26 NaHCO3, 20 glucose, 5 pyruvate, 1 CaCl2 and 2 MgCl2 and maintain at RT (95% O2, 5% CO2). This procedure allows for a good recovery and preservation of adult brain tissue after slicing. Compound action potentials (CAPs) were recorded in a recording chamber perfused with the extracellular solution at 2–3 mL/min at 34°C. For comparison between the three groups of interest, CAPs were always recorded in the same rostral *corpus callosum* region identified under the microscope by the size of lateral ventricles opposed to the SVZ (N = 5 mice per group). Compound action potentials (CAPs) were evoked by stimulating white matter fibers with a monopolar tungsten electrode while a recording electrode (glass pipette) was placed either in the lesion core or normal appearing white matter, according to the group (100 μs stimulations; Iso-Stim 01D, npi electronic GmbH, Tamm, Germany). Electrophysiological recordings were obtained using Multiclamp 700B, filtered at 4 kHz and digitized at 20 kHz. Digitized data were analyzed off-line using pClamp10.1 (Molecular Devices).

## Statistical analysis

All analyses were performed using GraphPad Prism 5.00 software (Sans Diego, California). All values are expressed as mean ± SEM. Each data group was first subjected to D'Agostino and Pearson normality Test. Statistical significance was determined using two-tailed Student's test or non-parametric Mann-Whitney test as indicated in figure legends. Multiple group comparisons were done using One or Two-Way ANOVA according experimental design or Kruskal-Wallis test (followed by a Dunn's multiple comparison post-hoc test). Quantifications were carried out from at least two or three independent experiments. Sample size (n) is mentioned in every figure legends. No statistical methods were used to predetermine sample sizes, but our sample sizes are similar those generally employed in the field. Variance was estimated for most results and no significant difference was found between control and experimental groups. The p-value $p<0.05$ was considered significant. *P* values, *t*-distributions and degrees of freedom (df) are given in figure legends.

## Study approval

All experimental procedures involving mice in our study were approved by the French Ministry of Agriculture (authorization number, 01169.02).

## Acknowledgements

We thank JL Thomas for helpful comments on the manuscript and CF Calvo for help with the neurosphere culture. The excellent animal care by Stephane Sosinsky and Fabien Uridat is gratefully acknowledged. The 'Plateforme d'Imagerie Cellulaire de la Pitié-Salpêtrière at ICM' provided excellent imaging advice. This work was supported by the EU FP7 contract Switchbox (Grant n° 259772), the EU H2020 contract Thyrage (Grant n°666869), the AFM; ANR grants Thrast and OLGA; CNRS; PRES/CUE Sorbonne Universités UPMC Univ Paris 06, INSERM; the program 'Investissements d'Avenir' ANR-10-IAIHU-06, NeurATRIS, the National Brain Research and Lendület Programs of Hungary. FCO is recipient of a post-doctoral fellowship from FRM and ARSEP and supported by FONDECYT INICIACION #11160616. MCA team is supported by FRM («Equipe FRM DEQ20150331681»), Idex-USPC and ANR grants.

## Additional information

### Funding

| Funder | Author |
| --- | --- |
| Association Française contre les Myopathies | Sylvie Remaud Barbara Demeneix |
| European Commission | Barbara Demeneix |
| Agence Nationale de la Recherche | Sylvie Remaud Bernard Zalc Barbara Demeneix |

| Fondation pour la Recherche Médicale | Maria Cecilia Angulo |

The funders had no role in study design, data collection and interpretation, or the decision to submit the work for publication.

## Author contributions

Sylvie Remaud, Conceptualization, Data curation, Formal analysis, Supervision, Funding acquisition, Validation, Visualization, Methodology, Writing—original draft; Fernando C Ortiz, Data curation, Formal analysis, Funding acquisition, Methodology, F.C.O carried out the electrophysiology experiments (compound action potentials measurements); Marine Perret-Jeanneret, Data curation, Formal analysis, Methodology; Marie-Stéphane Aigrot, Zsuzsanna Kvárta-Papp, Dominique Langui, Data curation, Formal analysis; Jean-David Gothié, Data curation; Csaba Fekete, Conceptualization, Data curation, Methodology; Balázs Gereben, Conceptualization, Data curation, Formal analysis; Catherine Lubetzki, Conceptualization; Maria Cecilia Angulo, Formal analysis, Supervision, Validation, Methodology; Bernard Zalc, Conceptualization, Supervision, Validation, Methodology, Writing—original draft; Barbara Demeneix, Conceptualization, Resources, Supervision, Funding acquisition, Writing—original draft, Project administration

## Author ORCIDs

Sylvie Remaud (iD) http://orcid.org/0000-0002-5490-6129
Barbara Demeneix (iD) http://orcid.org/0000-0003-4544-971X

## Ethics

Animal experimentation: All experimental procedures involving mice in our study were approved by the French Ministry of Agriculture (authorization number, 01169.02).

## Decision letter and Author response

Decision letter https://doi.org/10.7554/eLife.29996.020
Author response https://doi.org/10.7554/eLife.29996.021

## Additional files

### Supplementary files

• Transparent reporting form
DOI: https://doi.org/10.7554/eLife.29996.019

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
