## [Decision Letter]

[Editors’ note: a previous version of this study was rejected after peer review, but the authors submitted for reconsideration. The first decision letter after peer review is shown below.]

Thank you for submitting your work entitled "Transient hypothyroidism favors oligodendrocyte generation providing functional brain remyelination in vivo" for consideration by *eLife*. Your article has been reviewed by two peer reviewers, and the evaluation has been overseen by a Reviewing Editor and a Senior Editor. The reviewers have opted to remain anonymous.

Our decision has been reached after consultation between the reviewers and receiving additional comments from members of the *eLife* editorial board. Based on these discussions and the individual reviews, we regret to inform you that your work can not be considered further for publication in *eLife*.

As you will read below, the referees find your work in principle interesting and agree that transient hypothyroidism followed by T3 pulses comprise a novel way to approach the role of thyroid hormone role in myelination in vivo. However, there were too many concerns that the data presented do not justify the claims. There remained ambiguity around the origin of the cells contributing to remyelination and the many interesting and somewhat unexpected aspects of this paper do not "hang together as a whole to make an entirely compelling study" (as one referee put it). Referees were in particular missing direct evidence that the thicker myelin is mediated by newly generated cells from the SVC. The claims regarding translational potential appeared overstated.

The referees felt that it is not possible to solve the shortcomings of the present paper by reasonable revision experiments within a two month timeframe. A new submission with all points considered might receive more favourable comments. However, unlike a revision this would lead a new editorial assessment, most likely with different referees.

*Reviewer #1:*

The manuscript by Remaud, S. et al., seeks to show that:

1) A transient hypothyroid window (T_3_-/-), induced using MMI (0.1%)+NaClO4(0.5%), followed by 3 T_3_/T_4_ pulses, is sufficient to induce SVZ oligodendrogenesis. These SVZ-derived OPCs then migrate to the corpus callosum. Following toxic demyelinating injury (Cuprizone), SVZ-OPCs migrate into the corpus callosum to repopulate lost myelin.

2) These newly derived SVZ-OPCs are sufficient to functionally remyelinate the corpus callosum.

3) T_3_, and its endogenous receptor TRa1, are neurogenic. EGFR is gliogenic. T_3_ is sufficient to repress Egfr transcription in the adult SVZ.

4) TRa1 and EGFR segregate asymmetrically during mitosis, forming crescents on opposing poles during prophase. This asymmetric division within *Sox2Sox2*^+^ cycling cells is responsible for oligodendroglial or neuronal fate.

5) T_3_ bioavailability is determined by the opposing actions of the deiodinases Dio2 and Dio3. Dio3-the inactivating deiodinase, protects EGFR+ positive cells from the potential effects of T_3_ availability. Dio3 is sufficient to modulate neural stem cell fate distributions.

Review Summary:

The idea of focusing on transient hypothyroidism before T_3_ pulses is a novel, innovative way to approach thyroid hormone's role in re/myelination. However, while the authors show correlational data, there are major concerns with whether the data presented justify the ultimate claims. The authors have not concretely proven that T_3_ pulses following the transient hypothyroid window are not stimulating endogenous pOPC differentiation in the corpus callosum, accounting for the lateral localization of oligodendrocytes.

1) In Figure 1, the authors sought to prove that a hypothyroid window followed by T_3_ pulses was sufficient to stimulate SVZ oligodendrogenesis and migration to the corpus callosum, followed by differentiation and myelination. The authors claims are based on correlational data. In order to determine substantively that these OPCs were derived from the SVZ, the authors would need to fate map the cells to determine their origin, and eliminate the possibility that T_3_ pulses are not stimulating endogenous OPCs in the corpus callosum to differentiate.

2) Furthermore, to remove the potential contaminating effect of T_3_ on pOPC endogenous differentiation, I would like to see quantifications of cycling OPCs, number of myelinated axons, and g-ratios repeated across many models: control mice, euthyroid (no T_3_) mice, euthyroid mice, hypothyroid (no T_3_) mice, hypothyroid mice. This concern applies also to all tests of myelin functionality.

3) The authors suggest that TRa1 is excluded from cells expressing OLIG2 and SOX10, but that it is later turned on in cells expressing O4. This claim suggests that expression of TRa1 is developmentally regulated, thereby showing that expression of TRa1 is more intrinsically important than T_3_ pulses/general bioavailability of T_3_.

4) In the final figure, the authors claim that Dio3 is sufficient to modulate cell fate. However, the authors only show that the downregulation of Dio3 reduced cell proliferation by 50% and number of OLIG2+ progenitors by 30%. The authors do not show a subsequent increase in neuronal cells. This only shows fewer cells and does not prove that the plasmid is not simply killing cells. The authors would need to show a subsequent increase in neuronal cell fate in order to claim Dio3 expression is sufficient to modulate cell fate.

*Reviewer #2:*

This is a generally well-presented paper that reports some interesting and to some extent unexpected findings. Given the well-documented role of thyroid hormone in inducing oligodendrocyte differentiation from its progenitor (indeed TH is one the most reliable means of generation oligodendrocytes from OPs in tissue culture) it is unexpected that this hormone should have the opposite effect in guiding the differentiation of progenitors from the SEZ.

While the biology presented is of interest, the translational implications seem to me overstated. First, as the authors point out in the Introduction, the white matter adjacent to the SEZ tends to undergo efficient remyelination and is therefore not a major target for remyelination therapy. Second, since there is now reasonable and growing evidence that TH may have pro-remyelinating effects on OPs elsewhere in the CNS (that is beyond the very restricted region around the SEZ), inhibiting TH signalling to encourage remyelination close to the SEZ is likely to be detrimental to the regeneration of the much greater burden of demyelination throughout the rest of the CNS. In other words, the very small gain in repair close to the SEZ would likely be at the expense of limiting a much greater need for repair elsewhere – so why would you ever want to do it?

Introduction:

Is it really possible to identify mobilization of cells from the SEZ into adjacent tissue from MS post-mortem material? This seems to be inference rather than fact.

I would question whether SVZ-derived OPCs really produce 'thicker' myelin sheaths during remyelination. SVZ-OPCs remyelinate axons within the corpus callosum, which are amongst the smallest diameter axons in the CNS. It is well established that for these small diameter axons the myelin sheath thickness is the same for normally myelinated and remyelinated axons – it is only as axons get bigger that the thickness of the two diverge.

Results:

The authors should provide more details of how hypothyroidism is induced in the Results text.

Since Olig2 is expressed by all stages of the lineage it is not valid to use it as an OP marker and therefore neither is valid to state that there is a two-fold increase in OPs after 4 weeks of hypothyroidism.

The labelling for the graphs in Figure 1 need to be explicit indicating what the measurements are rather than what the inference is.

Re Figure 1'm sure it would be possible to provide some quantitative data on the MBP expression in the three groups.

Subsection “A T_3_-free window during demyelination favors functional remyelination”, first paragraph, the last sentence is entirely speculative and should be removed from the Results section.

The remyelination gradient that the authors refer to is not evident in Figure 1—figure supplement 1. In fact, it is difficult to glean any useful information from this figure.

At what stage were the g ratio measurements made? Is it possible that the EuCPZ animals just take longer to achieve full thickness? A later time point would resolve this.

The report that Olig2+ cells don't express TRb but O4+ cells do is very puzzling because all O4+ cells should also be Olig2+.

The staining in Figure 5 does not convince – it is very difficult to see co-localisation and the staining pattern for the PDGFRA-GFP is not what one would expect – it looks nuclear – is there an nls in the reporter construct?

[Editors’ note: what now follows is the decision letter after the authors submitted for further consideration.]

Thank you for resubmitting your work entitled "Transient hypothyroidism favors oligodendrocyte generation providing functional brain remyelination in vivo" for further consideration at *eLife*.

Your revised article has been favorably evaluated by Marianne Bronner (Senior Editor) and three reviewers, one of whom is a member of our Board of Reviewing Editors. The decision has been reached that the paper is acceptable for *eLife* provided that the following text changes are made to the manuscript.

1) One remaining issue, as pointed out by one referee, is whether the observed effects are spatially restricted or without further evidence generalizable to other white matter tracts ("it is likely to be true elsewhere in the brain"). The final revision of the manuscript should not suggest premature conclusions. Please rephrase.

2) Also the claim that scientist would believe 'demyelination is restricted to sensory motor systems' was not approved and should be dropped.

3) One referee writes: "…my point about mobilisation of cells in MS has been misunderstood. I am fully aware of the paper by Nait-Oumesmar and colleagues but my point is that the movement or mobilisation of cells is always an inference when one examines post-mortem MS tissue – it is never definitely proven and so should not be stated as such. It may well be the most likely interpretation but it is only that." Please adapt the text accordingly.

4) Please double-check, axis labelling (e.g.% of cycling cells is not the measurement: the measurement is% of olig2+ cells that are Edu+; also 'myelinated axons' is not a measurement, the label should be: number of remyelinated axons/square micron).

5) In the Abstract, please make clear what is introductory text and what refers to the experimental findings of the paper. Receptors (EGFR, TR) should not be termed "factors".

6) Please add a sentence, discussing whether the recovery from cuprizone was a transient effect or stable between treatment groups and thus of clinical relevance? Similarly, was this treatment sufficient to reduce axonal degeneration?

---

## [Author Response]

[Editors’ note: the author responses to the first round of peer review follow.]

Reviewer #1:[…] 1) In Figure 1, the authors sought to prove that a hypothyroid window followed by T_3_ pulses was sufficient to stimulate SVZ oligodendrogenesis and migration to the corpus callosum, followed by differentiation and myelination. The authors claims are based on correlational data. In order to determine substantively that these OPCs were derived from the SVZ, the authors would need to fate map the cells to determine their origin, and eliminate the possibility that T_3_ pulses are not stimulating endogenous OPCs in the corpus callosum to differentiate.

To analyze whether waves of MBP+/MOG+/OLIG2+ oligodendrocytes found laterally in the corpus callosum originate from endogenous adult pOPCs or from migrating progenitors newly generated within the adult SVZ, we specifically labeled SVZ-derived progenitors with a eGFP lentivirus. The stereotaxic injections of the viral vectors (1 µl of the lentiviral stock titered as 1.64E+09 TU/ml by FACS) were performed at 14d of demyelination. At 34d after starting the cuprizone diet, we observed not only GFP+ cells in the SVZ, but also groups of GFP+ cells that had migrated into the corpus callosum and into the striatum. We confirmed that more SVZ cells are recruited during a demyelinating insult in EuCPZ mice (3.6 ± 2,8 GFP+ cells per section, n=3) compared to control mice (2 ± 1,4 GFP+ cells per section, n=3). Hypothyroidism during cuprizone diet increases this recruitment even more. In HypoCPZ mice we found nearly 4 more GFP+ cells (11,8 ± 20,9 GFP+ cells per sections, n=3) and they display oligodendrocyte morphology (see Figure 1—figure supplement 2).

The question of whether T_3_ pulses are not stimulating pOPC to differentiate is addressed in the next question.

2) Furthermore, to remove the potential contaminating effect of T_3_ on pOPC endogenous differentiation, I would like to see quantifications of cycling OPCs, number of myelinated axons, and g-ratios repeated across many models: control mice, euthyroid (no T_3_) mice, euthyroid mice, hypothyroid (no T_3_) mice, hypothyroid mice. This concern applies also to all tests of myelin functionality.

This is indeed an interesting point. We have performed these control groups that just received NaCl after cuprizone treatment. Microtome sections were prepared to quantify myelinated axons in the corpus callosum, as has been done after T_3_ administration. Moreover, electron microscopy was performed to measure the g-ratio and thus to evaluate myelin thickness.

Figure 1—figure supplement 3 shows we observed that the numbers of myelinated axons in control and EuCPZ mice are not different after NaCl or T_3_ pulses during the recuperation phase. Moreover, in HypoCPZ mice, the density of myelinated axons is significantly increased compared to EuCPZ mice. This result shows that the hypothyroid window is sufficient to increase the density of myelinated axons. Accordingly, g-ratio measurements show that myelin thickness in HypoCPZ mice is also rescued. Thus, the T_3_-free window applied during demyelination, without the T_3_ pulses, is sufficient to promote generation of OPCs capable to remyelinate efficiently axons.

3) The authors suggest that TRa1 is excluded from cells expressing OLIG2 and SOX10, but that it is later turned on in cells expressing O4. This claim suggests that expression of TRa1 is developmentally regulated, thereby showing that expression of TRa1 is more intrinsically important than T_3_ pulses/general bioavailability of T_3_.

We are suggesting that during a transient period of commitment TRα1 is absent, but it will be developmentally induced, as TRs are required for oligodendrocyte differentiation (Barres et al., 1994). As to the effect of TRα1 expression in the absence of ligand (in this cellular context), this is a question that remains to be addressed.

4) In the final figure, the authors claim that Dio3 is sufficient to modulate cell fate. However, the authors only show that the downregulation of Dio3 reduced cell proliferation by 50% and number of OLIG2+ progenitors by 30%. The authors do not show a subsequent increase in neuronal cells. This only shows fewer cells and does not prove that the plasmid is not simply killing cells. The authors would need to show a subsequent increase in neuronal cell fate in order to claim Dio3 expression is sufficient to modulate cell fate.

We do not actually claim that Dio3 is sufficient to regulate cell fate decision. We just observed a decrease of both cell proliferation and oligodendrogenesis following a transient Dio3 knock-down.

We have changed the text accordingly to remove this ambiguity: “…down-regulation of *dio3* is sufficient to modify cell fate distribution” was replaced by “down-regulation of *dio3* is sufficient to modulate *oligodendrogenesis*”.

Reviewer #2:[…] While the biology presented is of interest, the translational implications seem to me overstated. First, as the authors point out in the Introduction, the white matter adjacent to the SEZ tends to undergo efficient remyelination and is therefore not a major target for remyelination therapy. Second, since there is now reasonable and growing evidence that TH may have pro-remyelinating effects on OPs elsewhere in the CNS (that is beyond the very restricted region around the SEZ), inhibiting TH signalling to encourage remyelination close to the SEZ is likely to be detrimental to the regeneration of the much greater burden of demyelination throughout the rest of the CNS. In other words, the very small gain in repair close to the SEZ would likely be at the expense of limiting a much greater need for repair elsewhere – so why would you ever want to do it?

The reviewer is right when (s)he asks why would hypothyroidism (transient) be a positive therapy for remyelinating MS plaques in humans, since many or most of those occur at some distance from the SVZ. Although our observation is limited to the corpus callosum, it is likely that it is also true elsewhere in the brain. These possibilities require investigation as in multiple sclerosis (MS) lesions are spread out in the brain and spinal cord. In the meantime even though our demonstration is limited to the corpus callosum, it is relevant to treatment of MS. Many people, even scientists, (though usually scientists who are not neurologists), think that demyelination effects are only seen in the sensory-motor systems (with the prototypical images of MS patients in wheelchair or with a white cane). However, it is important to recall that cognitive symptoms are also present in MS. In this respect lesions of the corpus callosum and of the prefrontal cortex are of primary concern. In terms of decision making such cognitive impairments may have crucial consequences. Of course remyelination with resident OPCs, i.e., thin myelin and short internode, may restore to some extent the speed of conduction, let’s say from a conduction block (demyelinated) to 45m/s (thin myelin) instead of 50m/s (thick myelin), but considering all our cognitive actions depending on rapid decision, even in our daily life, this may create a severe handicap. Such differences are always difficult to appreciate because interhemispheric lesions are rarely pure and their clinical consequences blurred by intrahemispheric lesions.

Furthermore, one should not limit one’s vision to general, whole body hypothyroidism. In fact a recent paper in Cell (Finan et al., 2016) showed the possibility of targeted thyroid hormone therapeutic approaches. Even if not immediately applicable, these possibilities have to be born in mind when discussing how our findings can stimulate new research avenues for many forms of neurodegenerative disease.

Introduction:Is it really possible to identify mobilization of cells from the SEZ into adjacent tissue from MS post-mortem material? This seems to be inference rather than fact.

We are referred to the published paper from Nait-Oumesmar et al., (2007) that has made this deduction from a number of autopsy specimen.

I would question whether SVZ-derived OPCs really produce 'thicker' myelin sheaths during remyelination. SVZ-OPCs remyelinate axons within the corpus callosum, which are amongst the smallest diameter axons in the CNS. It is well established that for these small diameter axons the myelin sheath thickness is the same for normally myelinated and remyelinated axons – it is only as axons get bigger that the thickness of the two diverge.

We beg to differ on this point. Perhaps, the situation can be better explained as follows.

On balance, the axons in the CC are small in diameter, but in their paper Xing et al., 2014 have shown with immunogold and EM that for a given axonal diameter axons in the corpus callosum remyelinated from SVZ originating OPC are thicker than those remyelinated from the parenchymal OPC (Figure 13 of their 2014 J Neurosci paper) – i.e., it doesn't seem to be an intrinsic property of small axons that they are all remyelinated to the same thickness. Similarly, Vittorio Gallo has shown in several papers that one can measure differences in gratio depending on the level of myelination in the corpus callosum (see for instance Hammond et al., Neuron. 2014). In our case, in addition to the similar EM evidence, we also provide significant differences in Compound Action Potential measurements.

Results:The authors should provide more details of how hypothyroidism is induced in the Results text.

As suggested this point has been clarified:

“Hypothyroidism was induced using MMI (0.1%) and NaClO4 (0.5%) during experimental demyelination induced by oral administration of cuprizone (CPZ) (Figure 1).”

Since Olig2 is expressed by all stages of the lineage it is not valid to use it as an OP marker and therefore neither is valid to state that there is a two-fold increase in OPs after 4 weeks of hypothyroidism.

The reviewer’s comment would be relevant if we were considering regions outside the neurogenic niche. However, regarding the SVZ niche, a major site of gliogenesis in the adult brain, Olig2 expression necessarily corresponds to early stages of determination. Moreover, we observed a strong co-expression of Olig2 and Sox10 in progenitors located within the adult SVZ (see Figure 3). In contrast, differentiation occurs in white matter close to the lateral ventricle, such as the corpus callosum or the striatum. Here Olig2 expression could be related to mature oligodendrocytes also expressing myelin proteins (MBP, MOG).

The labelling for the graphs in Figure 1 need to be explicit indicating what the measurements are rather than what the inference is.

We are not sure we understand the referee’s point here. The labelling is already precise as in each case we have added the analytical measure by which the data were quantified. In no case is the figure an inference but each represents data collected.

Re Figure 1'm sure it would be possible to provide some quantitative data on the MBP expression in the three groups.

As suggested, we quantified the MBP expression in each of the three groups (see Figure 1—figure supplement 1).

Subsection “A T_3_-free window during demyelination favors functional remyelination”, first paragraph, the last sentence is entirely speculative and should be removed from the Results section.

As suggested the last sentence was deleted.

The remyelination gradient that the authors refer to is not evident in Figure 1—figure supplement 1. In fact, it is difficult to glean any useful information from this figure.

We agree and have removed this graph G. Moreover, Figure 1—figure supplement 1 has been reorganized.

At what stage were the g ratio measurements made? Is it possible that the EuCPZ animals just take longer to achieve full thickness? A later time point would resolve this.

This point was addressed by the researcher that set up the cuprizone model (Ludwin, 1978 and 1979). Sam Ludwin gave 6 weeks for recovery and noted that: ‘The sheaths eventually reach a thickness approximately half that of normal development, with a disturbed relationship between myelin thickness and axon diameter’. So even extending the recovery period to 6 weeks does not allow full remyelination. Thus, showing that significantly greater functional recovery is found after transient hypothyroidism within one week post treatment adds to the clinical relevance of the finding.

The report that Olig2+ cells don't express TRb but O4+ cells do is very puzzling because all O4+ cells should also be Olig2+.

We show in vitro that TRa1 is expressed is some O4+ cells. We did not show in vitro immunohistochemistry against both Olig2 and TRa1. Olig2+ OP in the SVZ in vivo do not express TRa1 but we did not perform this double IHC in vitro. It is true that Olig2 expression persists in immature oligodendrocyte cells and thus we cannot exclude that TRa1 is detected in Olig2 immature oligodendrocytes in vitro.

The staining in Figure 5 does not convince – it is very difficult to see co-localisation and the staining pattern for the PDGFRA-GFP is not what one would expect – it looks nuclear – is there an nls in the reporter construct?

The mice PDGFRa^EGFP^ from The Jackson Laboratory express a H2B-eGFP fusion protein from the endogenous *Pdgfra* locus, hence the nuclear localisation (Figure 5).

[Editors' note: the author responses to the re-review follow.]

1) One remaining issue, as pointed out by one referee, is whether the observed effects are spatially restricted or without further evidence generalizable to other white matter tracts ("it is likely to be true elsewhere in the brain"). The final revision of the manuscript should not suggest premature conclusions. Please rephrase.

As suggested, this point has been clarified:

“A potential caveat might be that our observation is limited to the corpus callosum and that our findings cannot be extrapolated to other brain or spinal cord regions.”

2) Also the claim that scientist would believe 'demyelination is restricted to sensory motor systems' was not approved and should be dropped.

We agree and have removed this ambiguous statement: “lesions are not limited to motor control regions”.

3) One referee writes: "…my point about mobilisation of cells in MS has been misunderstood. I am fully aware of the paper by Nait-Oumesmar and colleagues but my point is that the movement or mobilisation of cells is always an inference when one examines post-mortem MS tissue – it is never definitely proven and so should not be stated as such. It may well be the most likely interpretation but it is only that." Please adapt the text accordingly.

The reviewer is right when (s)he asks whether it is possible to identify mobilization of cells from the SVZ into adjacent tissue from MS post-mortem tissues. We modified the text accordingly:

“…and *mouse* subventricular zone-derived oligodendrocyte progenitors that then generate newly formed OPCs locally (SVZ-OPCs) (Nait-Oumesmar et al., 2007).”

“The efficient remyelination capacity of SVZ–mobilized OPCs has been well-documented experimentally *in the mouse brain* (Nait-Oumesmar et al., 2007; Xing et al., 2014).”

“In this light, it is interesting to note that, *although the movement or mobilisation of cells can only be inferred when examining post-mortem tissue*, in brains of MS patients, early glial progenitors were observed more frequently in periventricular lesions as opposed to lesions further away from the ventricle (Nait-Oumesmar et al., 2007).”

4) Please double-check, axis labelling (e.g.% of cycling cells is not the measurement: the measurement is% of olig2+ cells that are Edu+; also 'myelinated axons' is not a measurement, the label should be: number of remyelinated axons/square micron)

As suggested, the labelling for the graphs in Figure 1 has been clarified.

5) In the Abstract, please make clear what is introductory text and what refers to the experimental findings of the paper. Receptors (EGFR, TR) should not be termed "factors".

As requested, the experimental findings were introduced accordingly:

“*Here we* report that a thyroid hormone (T_3_)-free window…”

Moreover, we clarified EGFR and TRα1 functions (Abstract) and removed the term “factor”: “…EGFR and TRα1, *expression of which favor glio- and neurogenesis, respectively*.”

6) Please add a sentence, discussing whether the recovery from cuprizone was a transient effect or stable between treatment groups and thus of clinical relevance?

This is indeed an interesting point. Accordingly, we have now added a statement: “Of note since animals were not examined for longer periods after stopping the cuprizone diet, we cannot ascertain the stability of the remyelination observed.”

Moreover, to avoid overstatement, we have deleted “…opening new therapeutic strategies for neurodegenerative diseases such as MS.”

Similarly, was this treatment sufficient to reduce axonal degeneration?

In light of the reviewer’s comment, the text has been clarified as follows: “In this respect, the recovery of CAPs suggests that remyelination in the cuprizone model preserves axon integrity as recently shown in an EAE model (Mei et al., 2016).”